# Mouse MRE11-RAD50-NBS1 is needed to start and extend meiotic DNA end resection

Soonjoung Kim [1,2] ✉, Shintaro Yamada [1,3], Tao Li [1], Claudia Canasto-Chibuque [1], Jun Hyun Kim[1], Marina Marcet-Ortega[1], Jiaqi Xu[1,4,6], Diana Y. Eng[1,7], Laura Feeney[1,8], John H. J. Petrini [1,4] & Scott Keeney [1,4,5] ✉

Nucleolytic resection of DNA ends is critical for homologous recombination, but its mechanism is not fully understood, particularly in mammalian meiosis. Here we examine roles of the conserved MRN complex (MRE11, RAD50, and NBS1) through genome-wide analysis of meiotic resection during spermatogenesis in mice with various MRN mutations, including several that cause chromosomal instability in humans. Meiotic DSBs form at elevated levels but remain unresected if *Mre11* is conditionally deleted, thus MRN is required for both resection initiation and regulation of DSB numbers. Resection lengths are reduced to varying degrees in MRN hypomorphs or if MRE11 nuclease activity is attenuated in a conditional nuclease-dead *Mre11* model. These findings unexpectedly establish that MRN is needed for longer-range extension of resection beyond that carried out by the orthologous proteins in budding yeast meiosis. Finally, resection defects are additively worsened by combining MRN and *Exo1* mutations, and mice that are unable to initiate resection or have greatly curtailed resection lengths experience catastrophic spermatogenic failure. Our results elucidate MRN roles in meiotic DSB end processing and establish the importance of resection for mammalian meiosis.

Homologous recombination during meiosis initiates with DNA double-strand breaks (DSBs) made by SPO11 protein, which remains covalently bound to DNA 5′ ends after strand breakage[1,2]. These DSBs must then be nucleolytically processed (resected) to generate the single-stranded DNA (ssDNA) needed for homology search and strand exchange (Fig. 1a)[3]. Despite its central role in recombination, meiotic DSB resection remains poorly understood, particularly in mammals.

The most detailed understanding of meiotic resection is currently for budding yeast, which uses a two-step DSB-processing mechanism (Fig. 1a)[3]. In the first step, the conserved Mre11-Rad50-Xrs2 (MRX) complex together with Sae2 (homologous to MRE11-RAD50-NBS1 (MRN) and CtIP in mouse) nicks the Spo11-bound strands using the endonuclease activity of Mre11 and degrades ssDNA towards the DSB using the 3′→5′ exonuclease activity of Mre11. We will refer to this as resection initiation. In the second step, the more processive 5′→3′ exonuclease activity of Exo1 further degrades ssDNA away from the DSB. This step has been referred to as long-range resection[4], although it should be noted that this is considerably shorter than the long-range resection that can occur in non-meiotic contexts[3].

[1]Molecular Biology Program, Memorial Sloan Kettering Cancer Center, New York, NY, USA. [2]Department of Microbiology and Immunology, Institute for Immunology and Immunological Diseases, Yonsei University College of Medicine, Seoul, Korea. [3]Department of Basic Medical Sciences, Tokyo Metropolitan Institute of Medical Science, Tokyo, Japan. [4]Weill Cornell Graduate School of Medical Sciences, New York, NY, USA. [5]Howard Hughes Medical Institute, Memorial Sloan Kettering Cancer Center, New York, NY, USA. [6]Present address: State Key Laboratory of Female Fertility Promotion, Center for Reproductive Medicine, Department of Obstetrics and Gynecology, Peking University Third Hospital, Beijing, China. [7]Present address: PackGene Biotech, Houston, TX, USA. [8]Present address: Translational Medicine, Oncology R&D, AstraZeneca, Barcelona, Spain. ✉e-mail: soonjk@yuhs.ac; s-keeney@ski.mskcc.org

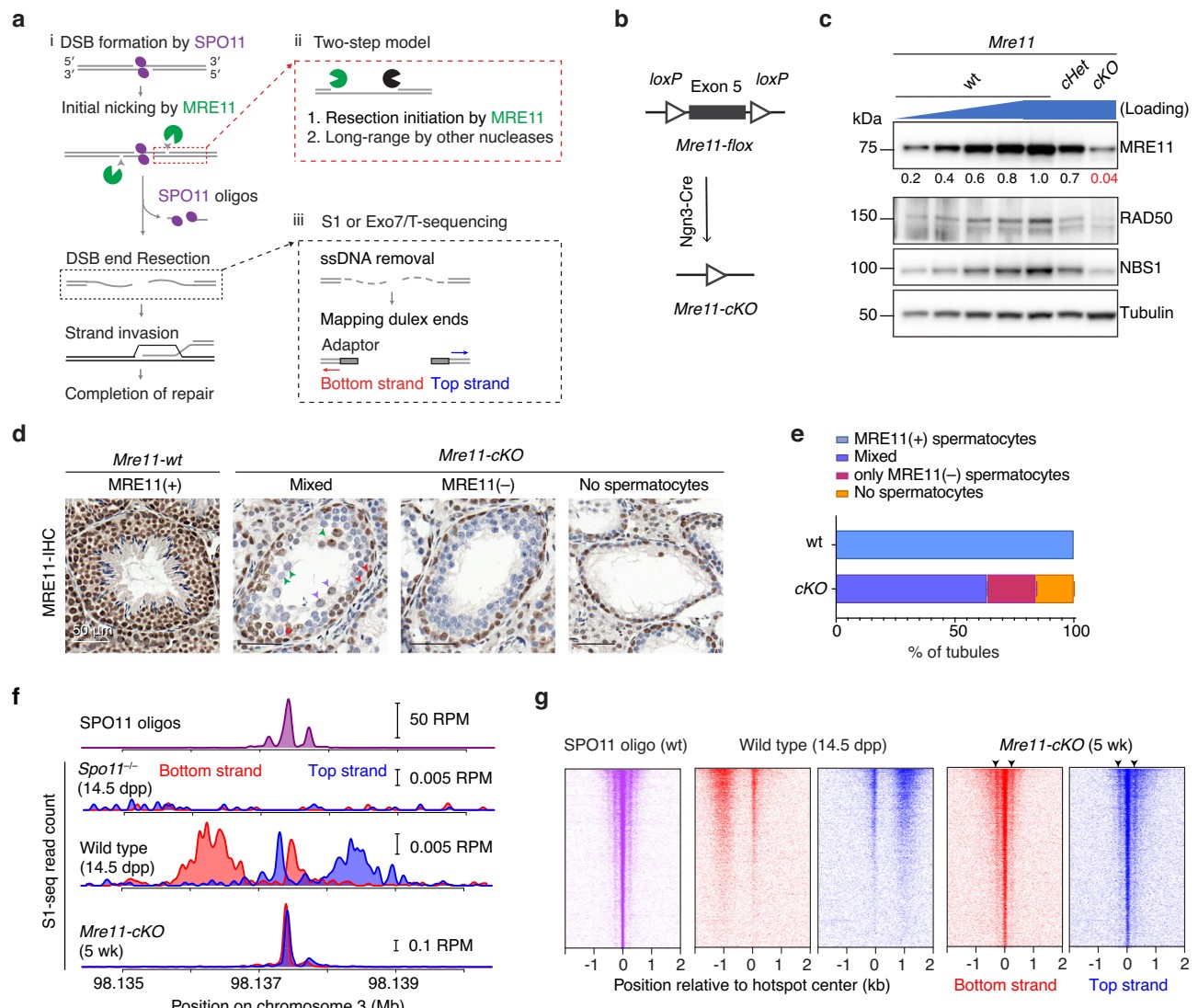

**Fig. 1 | MRE11 is required for resection initiation. a** Early meiotic recombination steps and sequencing strategy. i, SPO11 (magenta ellipses) cuts DNA via a covalent protein–DNA intermediate that is nicked (arrowheads) to give an entry point(s) for exonuclease(s). Bi-directional resection releases SPO11-oligo complexes and exposes ssDNA tails. ii, Two-step resection model: MRE11 hands off to other exonucleases. iii, Sequencing adapters are ligated to duplex ends after removal of ssDNA. Adapted from Fig. 1A of ref. [34] under a CC-BY 4.0 license (https://creativecommons.org/licenses/by/4.0/). **b** Conditional deletion of *Mre11*. **c** Reduced MRN protein in immunoblots of whole testis lysates from 5–7 wk old *Mre11-cKO* animals. Numbers below the MRE11 blot indicate amounts of a dilution series of wild-type (wt) lysate or band intensities for mutants relative to wild type, normalized to loading control (tubulin). **d** Seminiferous tubule sections at 5–7 wk stained for MRE11 (brown). In wild type, all cells except elongated spermatids stained positive for MRE11. In *Mre11-cKO*, tubules were categorized based on presence of MRE11-positive spermatocytes. Arrowheads, examples of MRE11-positive spermatogonia (red) or spermatocytes with (purple) or without (green)

MRE11 staining. **e** Quantification of tubule classes in (**d**). Number of animals tested: *n* = 1 for wild type and *n* = 2 for *Mre11-cKO* (mean ± range). **f** Strand-specific S1-seq [reads per million mapped reads (RPM)] at a representative DSB hotspot. Signals from three (wild type) or two (*Spo11*−/− and *Mre11-cKO*) biological replicate libraries were averaged and plotted after smoothing (151-bp Hann window). Choice of ages is explained in Methods. SPO11-oligo sequencing throughout is from our previous study[37]. The *y*-axis baseline for each plot is 0. **g** Stereotyped distribution of resection endpoints around DSB hotspots. Heatmaps (data in 40-bp bins) show strand-specific reads around SPO11-dependent DSB hotspots from wild type (*n* = 13,960 from our previous study[37]). Each hotspot is shown as a horizontal line, strongest at the top. Sequencing signals were locally normalized by dividing by the total signal in a 4001-bp window around each hotspot's center. Each hotspot thus has a total value of 1, to facilitate comparisons of spatial patterns between hotspots of different strengths. Arrowheads, the position of subsidiary peaks. Source data are provided as a Source data file.

The combined action of these nucleases generates long 3′-terminal ssDNA and releases Spo11 still covalently bound to short oligonucleotides (oligos)[2,4–13]. The average resection length in wild-type yeast is ~800 nt and this is reduced by more than half (to ~375 nt) in *exo1* nuclease-deficient mutants[4,12]. This residual meiotic resection in *exo1* mutants is similar in length to MRX/Sae2-dependent short-range resection in vegetative cells[3].

It is comparatively elusive how resection is executed in mammalian meiocytes. Similar to yeast, mouse DSB processing generates long

3′-terminal ssDNA (averaging ~1100 nt) and releases covalent SPO11-oligo complexes[8,14,15]. Unlike yeast, however, net resection length is decreased by only ~10% in mouse *Exo1* mutants, suggesting that EXO1 plays only a modest role in resection or is redundant with another nuclease(s)[14,15].

The contribution of MRN to resection has been difficult to address, in part because of experimental challenges due to the embryonic lethality of knockout mouse models lacking MRN subunits[16–18]. Conditional deletion of *Nbs1* in germ cells resulted in

reduced numbers of foci of RAD51 and other recombination proteins; this was interpreted to reflect a role for MRN in DSB processing, but resection per se was not directly assessed[19]. More recently, conditional *Rad50* deletion was found to cause an increased occurrence of unresected DSBs and to reduce resection tract lengths[20]. However, how MRN promotes resection remains unclear. Finally, it also remains unclear how resection length ties in with successful execution of meiotic recombination and thus with fertility.

We address these questions here through genome-wide analysis of meiotic DSB resection in mouse spermatocytes conditionally depleted of MRE11 or carrying various MRN mutations, including several that model hypomorphic genetic variants found in human chromosomal instability syndromes[21–28]. Our findings indicate that MRN is essential for resection initiation in mammals, but, surprisingly, it is also needed to generate most of the length of resection tracts that the yeast paradigm would instead ascribe to (Mre11-independent) long-range resection. Our findings further reveal how resection defects alter the binding of recombination factors to ssDNA and impinge on faithful execution of meiosis.

## Results

### MRE11 is required to initiate resection

To circumvent the lethality of an *Mre11* null mutation[17], we used a conditional deletion strategy combining a floxed *Mre11* allele[29] with *Ngn3-Cre*[30–32] (Fig. 1b). *Ngn3-Cre* is expressed in spermatogonia during the mitotic divisions prior to meiotic entry, but not in spermatogonial stem cells (SSCs, Supplementary Fig. 1a)[31,33]. We will refer to *Mre11^flox/−^ Ngn3-Cre^+^* as *Mre11-cKO* (for conditional knockout), *Mre11^flox/+^ Ngn3-Cre^−^* or *Mre11^flox/flox^ Ngn3-Cre^−^* as phenotypic wild type (wt), and *Mre11^flox/+^ Ngn3-Cre^+^* and *Mre11^flox/−^ Ngn3-Cre^−^* as cHet and Het, respectively.

MRE11 protein was reduced by about 20 fold in immunoblots of whole testis extracts from young adult *Mre11-cKO* males (5 wk old, Fig. 1c). We used immunohistochemistry on seminiferous tubule sections to determine which cell types retained MRE11 protein (Fig. 1d, e). Virtually all cells were positive for MRE11 in wild type, including the layers of spermatogenic cells and Sertoli cells within the tubules as well as interstitial somatic cells.

In *Mre11-cKO* mice, most tubules contained a basal layer(s) of spermatogonia and spermatocytes but lacked later-stage germ cells, and 16.1% of tubules lacked spermatocytes entirely (detailed further below) (Fig. 1d, e). Spermatogonia and interstitial cells remained uniformly MRE11-positive as expected, but spermatocytes, when present, either lacked detectable MRE11 or were a mix of MRE11-positive and -negative in the same tubule. These findings indicate that the *Mre11-flox* allele is recombined efficiently but incompletely by *Ngn3-Cre* expression in spermatogonia, resulting in a substantial population of spermatocytes entering meiosis with no or low MRE11 protein. As described before[29], RAD50 and NBS1 levels were also decreased when MRE11 was depleted (Fig. 1c).

We visualized resection endpoints genome-wide using S1-sequencing (S1-seq), in which genomic DNA is digested with the ssDNA-specific nuclease S1 and the blunted DNA ends are captured and sequenced (Fig. 1a)[4,15,34]. For some experiments, we used a combination of exonuclease VII and exonuclease T from *Escherichia coli* instead of nuclease S1 (Exo7/T-seq; based on END-seq)[14,35,36].

Most DSBs are formed in hotspots whose locations can be identified by sequencing of SPO11 oligos[37,38] (Fig. 1f, top). S1-seq and Exo7/T-seq yield *Spo11*-dependent reads distributed ~0.3–2 kb away from hotspot centers with defined polarity: top-strand reads from rightward-moving resection tracts and bottom-strand reads from leftward tracts[15] (Fig. 1a, f, g and Supplementary Fig. 1b, c). Both methods also generate reads close to hotspot centers (Fig. 1f, g and Supplementary Fig. 1b, c). This central signal is thought to arise from recombination intermediates, possibly D loops[14,15] (Supplementary Fig. 1d). Importantly, both S1-seq and Exo7/T-seq can also detect DSBs that have not been resected at all (Supplementary Fig. 1e)[4,14,15,39,40].

S1-seq and Exo7/T-seq profiles at hotspots were dramatically altered in young adult *Mre11-cKO* mice (Fig. 1f, g and Supplementary Fig. 1b, c). Virtually all signals accumulated at hotspot centers in a pattern strikingly similar to SPO11 oligos in wild type, with a strong central cluster and weaker flanking clusters to either side (Fig. 1f, g and Supplementary Fig. 1b, c). We conclude that DSBs remain unresected in the absence of MRE11. Our observations may also indicate that MRE11 is dispensable for DSB formation, but other interpretations are possible (Discussion).

### MRE11 controls DSB numbers and restrains double cutting

To summarize global patterns and to facilitate comparisons between genotypes, we plotted genome-wide profiles around DSB hotspots by co-orienting and averaging the sequencing signal from top and bottom strand reads (Fig. 2a). Compared to wild type, *Mre11-cKO* mice had elevated read counts at hotspots, with an increase well beyond the sample-to-sample variation in wild type (Fig. 2a, b). In response to DSBs, MRN complexes activate ataxia-telangiectasia mutated (ATM) protein kinase (Tel1 in budding yeast)[41–43]. During meiosis, ATM/Tel1 regulates the formation of DSBs via a negative feedback mechanism (reviewed in[44]) (Fig. 2c). We therefore surmised that the increased sequencing read count in *Mre11-cKO* mice reflects, at least in part, an increase in the number of DSBs per cell because of a loss of MRN-dependent ATM activation.

If so, we anticipated that *Mre11-cKO* mutants would also display another hallmark of ATM deficiency, namely, a high frequency of double cutting in which multiple SPO11 complexes cut the same DNA molecule in close proximity[45–47]. In support, the top-strand signal of the central S1-seq peak was shifted to the right of the SPO11-oligo peak, and the bottom-strand signal was shifted to the left (Fig. 2d). This pattern is distinct from the central signal derived from recombination intermediates in wild type (Supplementary Fig. 2a), but is as expected if multiple DSBs are formed on the same DNA molecules within hotspots because only the outermost DSB ends will be represented in the final sequencing library (Figs. 2c and Supplementary Fig. 1e). Double cutting may also explain why, unlike for SPO11 oligos in wild type, the subsidiary peak to the right of the central peak is stronger on average than the left-side subsidiary peak for top-strand reads, and vice versa for bottom-strand reads (Figs. 1g and 2d).

We further tested for double cutting by radiolabeling the 3′ ends of SPO11-oligo complexes immunoprecipitated from whole-testis extracts. This assay does not detect SPO11 if it is still bound to the high molecular weight DNA of unresected DSBs, but double cutting in budding yeast can generate Spo11-oligo complexes when Mre11 nuclease activity is compromised by *rad50S* or *sae2Δ* mutations[45,47] (Fig. 2c). We indeed observed SPO11-oligo complexes in *Mre11-cKO* mice, and these were at levels more than ten-fold greater on a per-testis basis than in wild-type or heterozygote controls (Fig. 2e and Supplementary Fig. 2b), comparable to *Atm^−/−^* mice[48].

The major band of SPO11-oligo complexes from *Mre11-cKO* had slower mobility on SDS-PAGE than controls (Figs. 2e and Supplementary Fig. 2b). We therefore determined the size distribution of radiolabeled SPO11 oligos on denaturing PAGE (Fig. 2f). The SPO11 oligos in wild type that arise from canonical resection initiation have a bimodal size distribution, with most falling in the ranges of ~15–27 or ~31–35 nucleotides long[8,37,48,49]. By contrast, SPO11 oligos that arise from double cutting are at least ~33 nt and range to much larger sizes, with a pronounced ~10-nt periodicity in lengths that is a diagnostic feature of SPO11 double cuts thought to reflect geometric constraints on the orientation of adjacent SPO11 dimers relative to the DNA[45,47,50]. Because *Atm^−/−^* spermatocytes retain the ability to initiate resection[14,15], they have a mix of SPO11 oligos arising from both resection and SPO11 double cutting[45,48] (Fig. 2f).

Similar to SPO11 oligos purified from *Atm^−/−^* mice compared side by side, those from *Mre11-cKO* mice displayed a major species migrating slightly slower than the 34-nt marker plus a ladder of slower

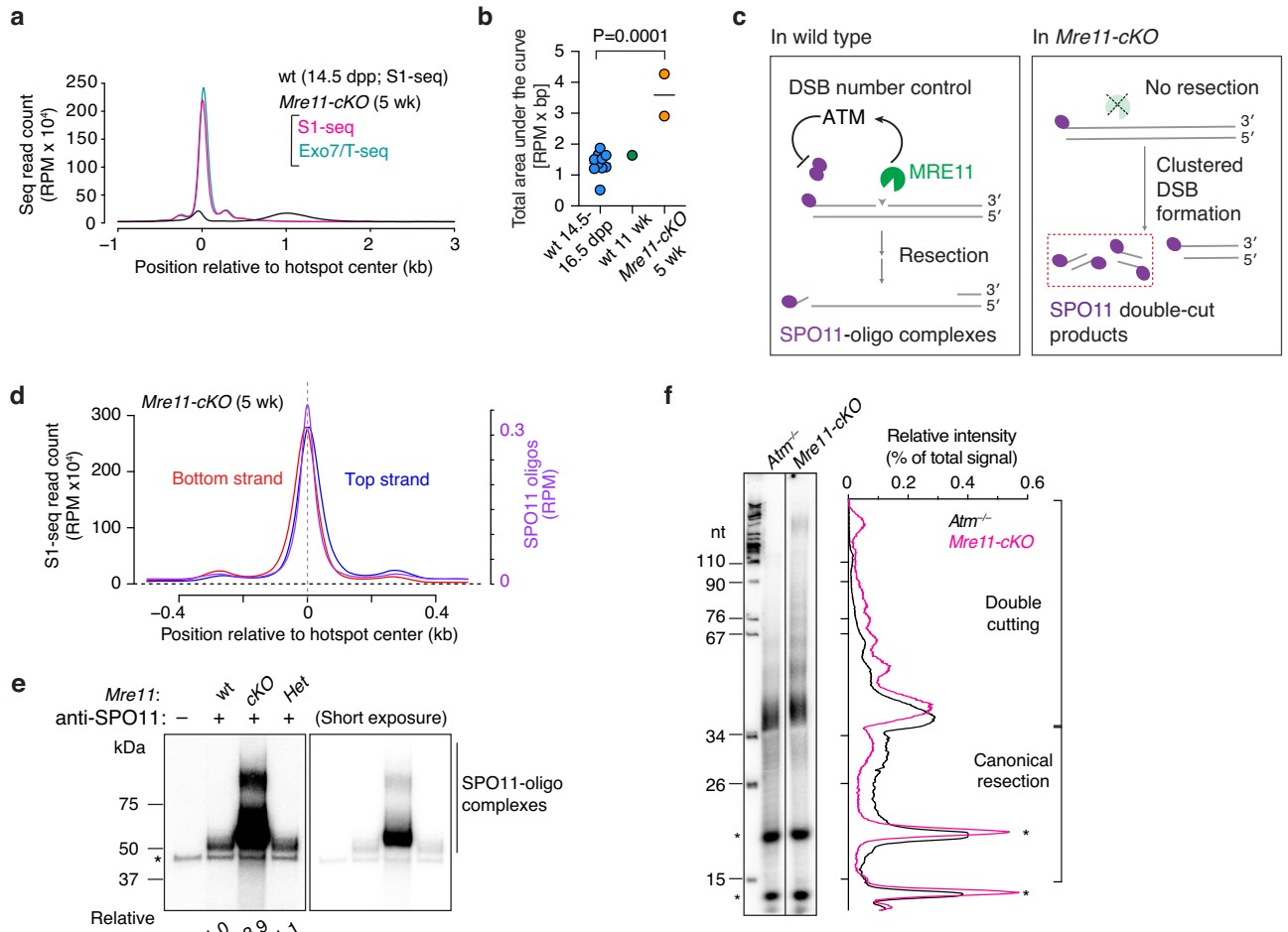

**Fig. 2 | MRE11 fosters DSB number control. a** Genome-average S1-seq and Exo7/T-seq at hotspots in 5 wk old *Mre11-cKO* animals. Bottom-strand reads were flipped and combined with top-strand reads, averaged, and smoothed (151-bp Hann window; average of two biological replicates). S1-seq in 14.5-dpp wild type is shown on the same scale (average of three biological replicates). **b** S1-seq signal for *Mre11-cKO* (n = 2) and wild type (n = 10 for juvenile, 1 for adult). Each point is the area under the genome-average curve (as in panel **a**) from −1500 to +2500 bp relative to the hotspot center for an individual replicate. The *P* value is from a Student's t test (two-sided). **c** Left, ATM activation by MRE11 and subsequent negative feedback regulation of SPO11. Right, absence of resection and increase in SPO11 double cuts in the absence of MRE11. **d** Offset top- and bottom-strand reads at hotspot centers in *Mre11-cKO*, consistent with double cutting. The plot shows the smoothed genome-average strand-specific profile close to hotspot centers (51-bp Hann window) with the SPO11-oligo profile for comparison. **e** Increased SPO11-oligo complexes with altered electrophoretic mobility in *Mre11-cKO*. The autoradiograph shows SPO11-oligo complexes immunoprecipitated from testis extracts from 5–6

wk old mice, radiolabeled with terminal transferase and [α-$^{32}$P]-dCTP, and separated by SDS-PAGE. The signal intensity of the region above the non-specific band, normalized to wild type, is indicated below. Asterisk: non-specific labeling. **f** SPO11 oligo lengths in *Mre11-cKO*. SPO11 oligos were purified from two mice of each genotype by immunoprecipitation and protease digestion, then radiolabeled with terminal transferase and [α-$^{32}$P]-GTP and separated by denaturing urea PAGE. Note that SPO11 oligos appear slightly larger than their true lengths because of nucleotide(s) added by terminal transferase and residual amino acid(s) not removed by protease. Both samples were run on the same gel, but the intensity of the *Mre11-cKO* signal was increased approximately threefold to facilitate comparison. Asterisks: non-specific species; nt: markers in nucleotides. Lane traces (background-subtracted and normalized to the total lane signal) are shown to the right. Expected migration positions of species arising from MRE11-dependent (canonical) resection and from SPO11 double cutting are indicated. Source data are provided as a Source data file.

migrating bands with ~10-nt periodicity. Interestingly, however, compared with *Atm*$^{-/-}$, the *Mre11-cKO* sample displayed substantially less labeled material migrating faster than the 34-nt marker (Fig. 2f). These findings together are consistent with the interpretation that most if not all of the SPO11-oligo complexes in *Mre11-cKO* spermatocytes arise from double cuts.

We also examined resection patterns in juvenile (14.5 dpp) *Mre11-cKO* mice. Immunoblotting of whole-testis extracts showed that MRE11 protein was reduced, but to a lesser degree than in adults ( ~ twofold lower than controls; Supplementary Fig. 2c). Similar to adults, these mice showed a greatly elevated central S1-seq signal at hotspots that had the spatial patterns diagnostic of unresected DSBs and double cutting (Supplementary Figs. 2d–f). Unlike adults, however, juveniles also had a substantial fraction of resected DSBs, although with a

resection length distribution that was markedly shorter (modal length 897 nt) than in wild type (modal length 1015 nt) (Supplementary Fig. 2f). It is formally possible that spermatocytes in the initial wave of meiosis in juveniles have an additional (MRE11-independent) resection initiation mechanism that is not available in adults. We consider it more likely, however, that the weaker phenotype in juveniles reflects differences in the timing of Cre-mediated excision relative to DSB formation and/or differences in the kinetics of MRE11 protein rundown after excision.

### Spermatogenesis failure in *Mre11*-deficient mice
Adult *Mre11-cKO* mice had significantly smaller testes than littermate wild-type or heterozygous controls (Fig. 3a, b). In seminiferous tubule sections, layers of spermatogonia and spermatocytes were present but

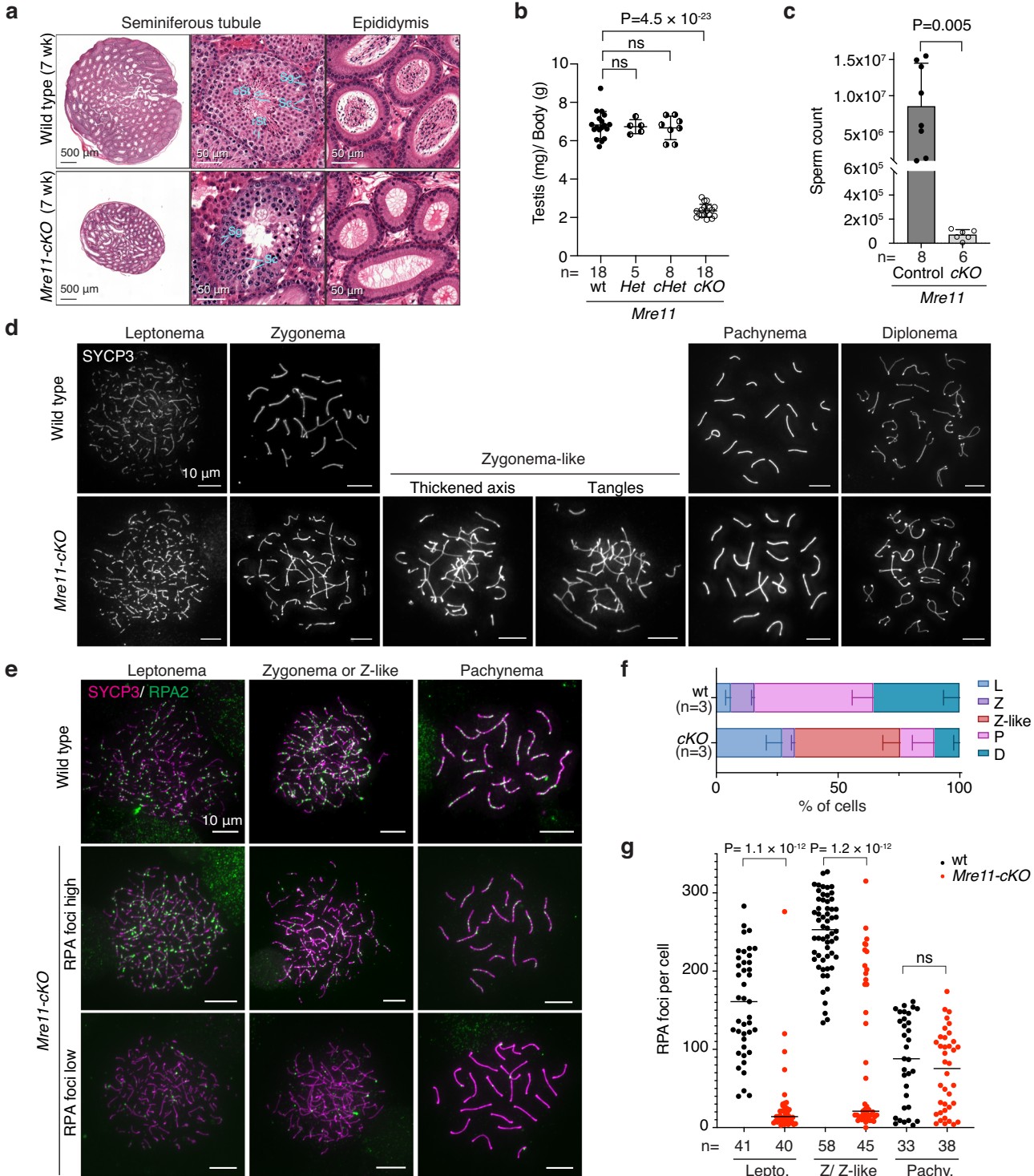

**Fig. 3 | Spermatogenesis defects in *Mre11*-deficient mice. a** Bouin's fixed testis and epididymis sections stained with hematoxylin and eosin (H&E). Sg: spermatogonia, Sc: spermatocyte, rSt: round spermatid, eSt: elongated spermatid. **b** Ratios of testis weight (mg per pair) to body weight (g). Each point represents one animal; error bars indicate mean ± SD. The results of Student's t tests (two-sided) are shown; ns, not significant (*P* > 0.05). **c** Sperm counts (mean ± SD; *P* value from Student's t test (two-sided)). **d** Representative spreads of wild-type or *Mre11-cKO* spermatocytes showing normal SYCP3 staining at the indicated stages or abnormal ("Zygonema-like") cells with thickened and/or tangled axes.

**e** Representative RPA2 staining on spermatocyte chromosome spreads. Examples are shown of *Mre11-cKO* cells with either low or high RPA2 focus counts. **f** Spermatocyte stages based on SYCP3 staining (L: leptonema, Z: zygonema, Z-like: zygonema-like, P: pachynema, D: diplonema). Error bars indicate mean ± SD from three mice of each genotype. **g** RPA2 focus numbers. The horizontal lines represent medians for the indicated number of cells counted from 2 mice of each genotype. The stage of each cell was determined from the SYCP3 signal. The results of two-tailed Mann-Whitney U tests are shown. Source data are provided as a Source data file.

postmeiotic cells were largely absent, sections of epididymis appeared empty, and epididymal sperm counts were greatly reduced (Fig. 3a, c). All *Mre11-cKO* tubules contained one or more layers of cells that stained positive for the mouse VASA homolog DDX4 (Supplementary Fig. 3a), which is expressed from spermatogonia to the round spermatid stage[51]. Tubules with apoptotic cells were also increased in frequency (Supplementary Figs. 3b, c). These *Mre11-cKO* phenotypes remained consistent to at least 16 wk of age (the oldest tested) (Supplementary Fig. 3d). These results, together with MRE11 staining (Fig. 1d), support the interpretation that *Mre11-cKO* mice stably maintain an MRE11-positive SSC population that can initiate multiple waves of spermatogenesis, but most spermatocytes in each wave are then eliminated by apoptosis. We conclude that MRE11 protein is essential for male fertility in mice, as are NBS1 and RAD50[19,20].

To monitor chromosome dynamics and progression through prophase I, we immunostained spermatocyte chromosome spreads for the axis protein SYCP3, with or without costaining for the RPA2 subunit of the ssDNA-binding protein RPA. In wild type, SYCP3 forms short lines in leptonema as axes begin to form; the axes elongate and synapse with homologous partners to form stretches of synaptonemal complex (SC) in zygonema; SC spans the length of each pair of autosomes in pachynema; and then SC disassembles in diplonema (Figs. 3d and Supplementary Fig. 3e). RPA2 forms large numbers of foci on resected DSBs in leptonema and zygonema, then these foci decrease in number over the course of pachynema as DSB repair is completed (Fig. 3e)[52].

Spermatocytes in *Mre11-cKO* mice deviated substantially from the normal patterns. We observed cells with short SYCP3 axes and no SC, similar to normal leptonema but representing a higher fraction of total spermatocytes than in wild type (Fig. 3d, f and Supplementary Fig. 3e). Most cells had very few RPA2 foci (Fig. 3e, g). We also observed cells with axes of various lengths along with SC, but most of these differed from normal zygotene cells in having only short stretches of staining for SC central region component SYCP1 and/or tangles of axes (Supplementary Fig. 3e). These "zygotene-like" cells were the most abundant class in *Mre11-cKO* (Fig. 3f) and most were again highly depleted for RPA2 foci compared to wild type (Fig. 3e, g). The deficit of RPA2 foci corroborates the resection defect seen in S1-seq. These results also show, not surprisingly, that unresected DSBs cannot support homologous pairing and synapsis.

In addition to these aberrant cells, there were also morphologically normal pachytene and diplotene cells, but much fewer than in wild type (Figs. 3d, f and Supplementary Fig. 3e). RPA2 focus numbers in the pachytene cells were indistinguishable from wild type (Fig. 3g). We also observed a small fraction of leptotene and zygotene cells that had RPA2 focus numbers in the normal range for these stages (Fig. 3e, g). We infer that these more normal-looking cells correspond to the MRE11-positive subpopulation of spermatocytes (Fig. 1d), perhaps cells that escaped Cre-mediated excision. If so, these escapers appear to be too small a fraction of the total leptotene and zygotene cells (Fig. 3f) to contribute an appreciable resection signal in population-average sequencing maps (Figs. 1f, g, and 2a).

## Nuclease-dead MRE11 reveals an unexpected role in extending resection lengths

To test whether MRE11 nuclease activity is required for meiotic resection, we examined mice with the invariant active-site residue His-129 substituted to asparagine[29]. Human MRE11 with this mutation assembles MRN complexes and activates ATM, but lacks nuclease activity in vitro[53,54]. *Mre11-H129N* is cell-lethal in chicken and human cells[55] and is embryonically lethal in mice[29], but the phenotype in mammalian meiosis has not been addressed. To circumvent the lethality, we generated *Mre11^H129N/flox* mice carrying *Ngn3-Cre* (hereafter *Mre11-cHN*) (Fig. 4a). Total MRE11 protein levels were normal in testis extracts from *Mre11-cHN* mice (Fig. 4b), consistent with the previous

demonstration that the mutant mouse protein is stable and assembles MRN complexes normally[29].

Because the equivalent budding and fission yeast mutants form meiotic DSBs but cannot resect them[7,56], we expected that *Mre11-cHN* mice would accumulate only unresected DSBs if the wild-type MRE11 protein expressed from the floxed allele were sufficiently depleted (Supplementary Fig. 4a, left). On the other hand, if some wild-type MRE11 protein persisted long enough to support resection initiation at a subset of DSBs in each cell, or at all DSBs in a subset of escapee cells, we instead expected to observe that some or all of the DSBs would be fully resected (Supplementary Fig. 4a, right). The latter prediction was premised on the idea that the long-range nuclease(s) should be able to complete resection normally as long as MRE11 endonuclease had incised the DNA. Similar logic would apply to interpretation of the resection that occurs in juvenile *Mre11-cKO* mice.

Surprisingly, however, Exo7/T-seq patterns in young adult mice matched neither of these predictions when considering either individual hotspots (Figs. 4c and Supplementary Fig. 4b) or genome-average profiles (Fig. 4d and Supplementary Fig. 4c). We note two main findings. First, there was a modest increase in the relative amount of the sequencing signal at hotspot centers (Fig. 4d), suggesting that some unresected DSBs are present. Consistent with this interpretation, we also observed a slight increase in the sequencing signal at hotspot centers on the nonhomologous parts of the X and Y chromosomes (Supplementary Fig. 4d). Central S1-seq or Exo7/T-seq signal from recombination intermediates does not appear at X and Y hotspots in wild type, but it can be detected when there are unresected DSBs, such as in *Atm^-/-* mutants[14,15] or *Mre11-cKO* (Supplementary Fig. 4e).

To further test for unresected DSBs, we performed chromatin immunoprecipitation (ChIP-seq) for MRE11, which was previously demonstrated to accumulate at hotspots when unresected DSBs are present in *Atm^-/-* mutants, but not in wild type where all DSBs are resected[14]. Indeed, we observed a clear MRE11 ChIP-seq signal at hotspots in *Mre11-cHN* mice, albeit substantially less than in *Atm^-/-* (Figs. 4e and Supplementary Fig. 4f, g). Taken together, these results suggest that a small fraction of DSBs remains unresected in *Mre11-cHN* mice. If so, however, this resection initiation defect is very weak compared to *Atm^-/-* and especially *Mre11-cKO*, perhaps reflecting insufficient elimination of wild-type MRE11 protein in this conditional model (see Discussion).

The second and more surprising finding was that resection tracts were substantially shorter than normal in *Mre11-cHN*, averaging only 672 nt (Fig. 4d). This defect was reproducible and stable, remaining the same with increasing age up to 11 wk (the oldest age tested; Supplementary Fig. 4h). We also observed a slight resection defect in mice heterozygous for *Mre11-H129N* (*Mre11^H129N/wt* with *Ngn3-Cre*). Resection tracts in these mice averaged 1040 nt (94% of wild type) (Fig. 4d), suggesting a weak dominant-negative effect of MRE11-H129N protein. Observing shorter average resection lengths suggests that MRE11 nuclease activity contributes not just to resection initiation, but also to the extension of resection tracts beyond the lengths seen for MRX-dependent short-range resection in yeast.

Unlike in *Atm^-/-* and *Mre11-cKO*, the amount of SPO11-oligo complexes was not increased in *Mre11-cHN* (Figs. 4f and Supplementary Fig. 4i), consistent with MRE11-HN protein being competent to activate ATM/Tel1 in both yeast and mouse[29,54,57,58]. Instead, the yield on a per-testis basis was reduced by about half compared to wild type. The total Exo7/T-seq signal recovered at hotspots was also diminished in *Mre11-cHN* (Supplementary Fig. 4j). These decreases may reflect reduced DSB frequency and/or loss of signal because of spermatocyte apoptosis (described below). We also observed a reproducible shift in the migration of SPO11-oligo complexes on SDS-PAGE gels, with faster migrating species appearing to be more depleted than slower migrating ones (Fig. 4g, h). We posit that the presence of catalytically inactive MRE11 protein sometimes occludes potential nicking sites

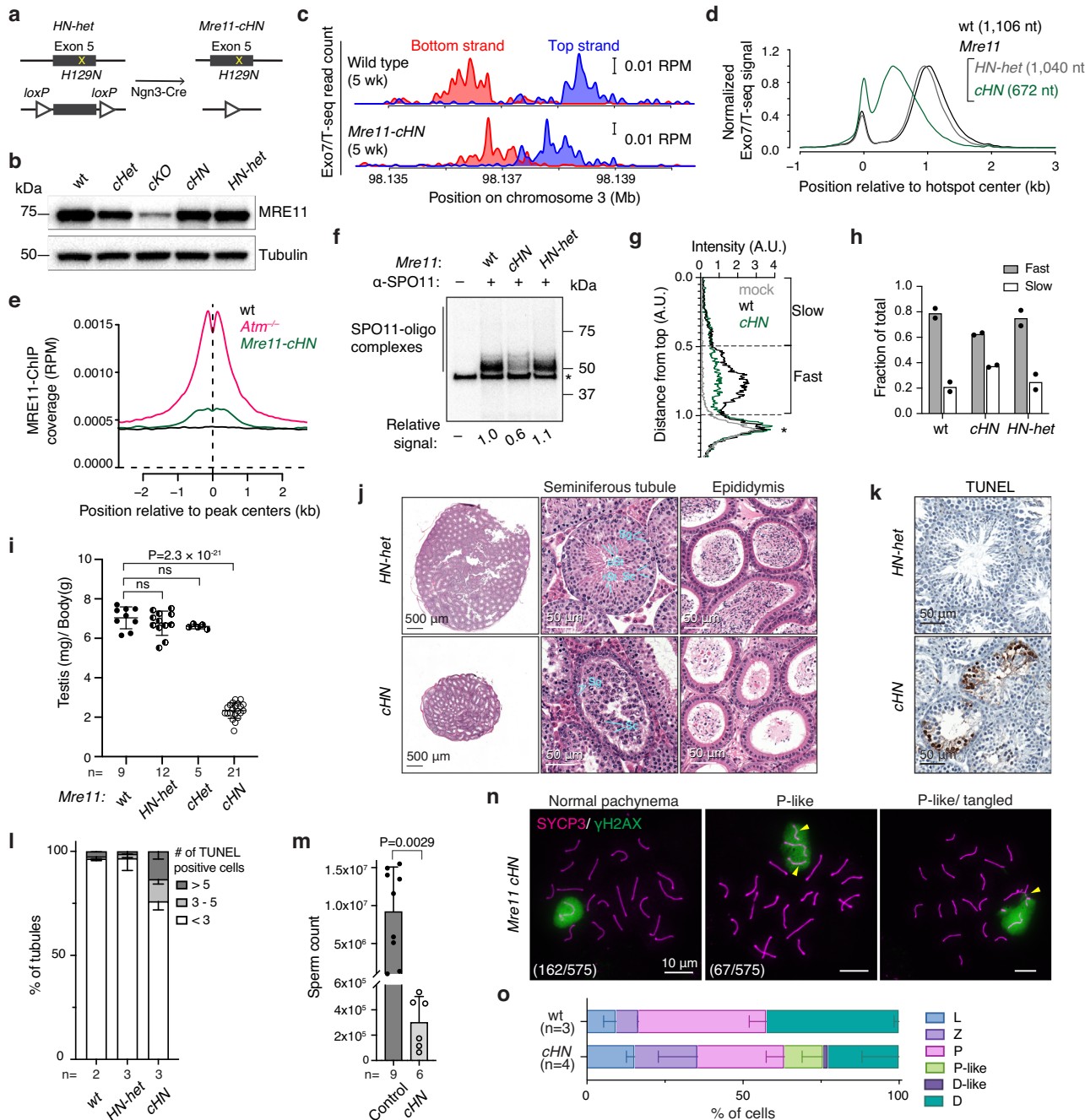

**Fig. 4 | Defects in resection and spermatogenesis in mice expressing nuclease-dead MRE11. a** Conditional *Mre11-H129N* strategy combining *Ngn3-Cre* with compound heterozygous floxed and missense *Mre11* exon 5 alleles. **b** Normal MRE11 protein levels of whole testis lysates from 5–7 wk old *Mre11-cHN* mice. Here, the phenotypically wild-type control (wt) was *Mre11^wt/flox^ Ngn3-Cre^−^* and the heterozygous HN strain (*HN-Het*) was *Mre11^H129N/flox^ Ngn3-Cre^−^*. **c** Exo7/T-seq at the hotspot shown in Fig. 1f. **d** Genome-average Exo7/T-seq at hotspots (5-wk animals). Top and bottom strand reads were co-oriented, averaged, and smoothed (151-bp Hann window), then internally normalized by setting the height of the resection peak to 1. Normalization facilitates comparison of shapes of resection profiles separate from sample-to-sample variation in read frequencies[34]. Non-normalized plots are in Supplementary Fig. 4c. n = 2 for wt and *Mre11-cHN*, *n* = 1 for *HN-Het*. **e** MRE11 accumulation at hotspots in *Mre11-cHN*. MRE11 ChIP-seq signal was averaged around hotspots for *Mre11-cHN* (*n* = 3), wt (*n* = 2), and *Atm^−/−^* (*n* = 1). **f, g** Fewer SPO11-oligo complexes in *Mre11-cHN*. Autoradiograph of radiolabeled complexes immunoprecipitated from testis extracts (**f**; signal intensity excluding the non-

specific band, relative to wild type, indicated below). Lane profiles (**g**) depicting fast and slow migrating species as a fraction of total signal are shown. Dashed lines represent the boundary of each species. Asterisk, non-specific labeling. **h** The bar graph summarizing results from two biological replicates. **i** Reduced testis sizes (mean ± SD). Results of Student's t tests (two-sided) are shown. **j** Seminiferous tubule and epididymis sections (7 wk old; cell type labels as in Fig. 3a). **k, l** Increased spermatocyte apoptosis in *Mre11-cHN* mice (7 wk). TUNEL staining (**k**) and quantification (**l**) are shown. Error bars in (**l**) indicate mean and range. Wild type reproduced from Supplementary Fig. 3c. **m** Sperm counts (mean ± SD; *P* value from Student's t test (two-sided)). Control combines wt, *HN-het*, and *cHet*. **n** *Mre11-cHN* spermatocyte spreads showing either normally synapsed autosomes and normal γH2AX-positive sex body (left) or pachytene-like cells with unpaired or tangled autosomes (middle and right; arrowheads, γH2AX-positive domains). Numbers indicate pachytene(-like) cells observed and SYCP3-positive cells counted. **o** Spermatocyte stages (mean ± SD). Source data are provided as a Source data file.

closer to SPO11, biasing the population of SPO11 oligos generated by the residual wild-type MRE11 protein to larger sizes than normal.

We were unable to assess a role for CtIP—a cofactor needed for MRN nuclease activity[3]—because conditional deletion of *Ctip* eliminated spermatogonia prior to meiotic entry (Supplementary Figs. 4k–n). We also tested *Rad50^{S/S}* mice but found normal resection (Supplementary Fig. 4o), likely because this *Rad50-K22M* mutation was based on a relatively weak yeast allele because mimics of stronger alleles are cell-lethal[23]. Thus, available mouse mutants are unable to definitively test the requirement for MRN nuclease in resection initiation.

### Spermatogenesis defects in mice with nuclease-dead MRE11

*Mre11-cHN* mice had pronounced defects in meiotic progression. Their testes were considerably smaller than those of control littermates (Fig. 4i); seminiferous tubules contained spermatogonia and early stage spermatocytes but were depleted of spermatids and had an increased frequency of spermatocyte apoptosis (Fig. 4j–l); and epididymal sperm counts were greatly diminished (Fig. 4j, m). The rare residual sperm detected may have arisen from escapers in which Cre-mediated excision failed and/or from a subset of cells in which wild-type MRE11 protein persisted long enough after excision to support successful meiosis.

When prophase I stages were evaluated by staining spreads for SYCP3 and phosphorylated H2AX (γH2AX), we observed a subset of pachytene-like cells with unpaired or tangled chromosomes (Fig. 4n, o), indicative of sporadic synaptic failure. We also observed an increase in the proportions of leptotene and zygotene cells relative to later stages (pachytene, pachytene-like, and diplotene) (Fig. 4o). This increase may reflect a delay in completing synapsis, preferential apoptotic elimination of cells in later prophase I, or both.

### Reduced resection lengths in MRN hypomorphic mutants

We further tested for MRN roles in meiotic resection using mice homozygous for hypomorphic mutations modeled on human disease alleles[59] (Fig. 5a). *Mre11-ATLD1* is a nonsense mutation near the 3′ end of the coding sequence, recapitulating a mutation found in ataxia telangiectasia-like disorder[21]. *Nbs1ΔB*, which models Nijmegen breakage syndrome, produces an NBS1 protein lacking the N-terminal FHA and BRCT domains[26]. *Mre11-ATLD1* and *Nbs1ΔB* cause intra-S and G2/M checkpoint defects, DNA damage sensitivity, chromosomal instability, and reduced ATM activity in cultured cells and they cause subfertility in mice, particularly in females[21,26,60]. The mutations also reduce protein stability in cells[21,26] and MRN nuclease activity in vitro[61], but they have not previously been linked to overt resection defects.

We also examined homozygous *Rad50-D69Y* and *Rad50-L1237F* mutants, which model recurrent mutations in human cancer affecting residues in the Walker A and B ATPase motifs, respectively[62,63] (Fig. 5a). Studies in *S. cerevisiae* and cultured mouse cells indicated that the human *RAD50-L1240F* mutant (*L1237F* in mouse) is a separation-of-function allele that is largely proficient in DNA repair, but defective in the activation of Tel1/ATM[62]. Similar outcomes were observed for modeling of human *RAD50-D69Y* in *S. cerevisiae*[63].

*Rad50-L1237F* and *Rad50-D69Y* mice were derived by CRISPR-mediated mutagenesis. MEFs from *Rad50-D69Y* mice displayed reduced ionizing radiation (IR)-provoked phosphorylation of KAP1 (an ATM target[64]) (Supplementary Fig. 5a); elevated mitotic indices after IR, consistent with defects in the ATM-dependent G2/M checkpoint (Supplementary Fig. 5b); and hypersensitivity to camptothecin when ATR was inhibited, as expected if compromised ATM activation cannot compensate for loss of ATR activity (Supplementary Fig. 5c). These findings confirm the separation of function predicted from prior studies in *S. cerevisiae*[63], and verify that *Rad50-D69Y*, like *Rad50-1237F*[62], is defective for ATM activation.

S1-seq maps from juvenile males demonstrated resection defects in all of these MRN mutants, demonstrating a striking allelic series

(Fig. 5b–d). These findings reinforce the conclusion that the MRN complex must be fully functional to support normal resection lengths, i.e., that MRN roles in DSB processing are not limited just to resection initiation close (within a few hundred bp) to the DSB. There was only a slight increase in central S1-seq signal at hotspot centers on the sex chromosomes in *Nbs1ΔB*, suggestive of a small number of unresected DSBs, but little if any evidence for this in the other mutants (Supplementary Fig. 5d).

Spermatogenesis in *Nbs1ΔB* and *Mre11-ATLD1* mutant males was previously shown to be nearly normal, with only mildly delayed temporal progression and persistent RAD51 localization that was attributed, at least in part, to reduced functionality of these MRN hypomorphs in meiotic DSB repair[21,60]. The substantially shorter resection lengths we now observe in these mutants may contribute to this apparent DSB repair delay. Spermatogenesis also appeared to be largely normal in *Rad50-L1237F* and *-D69Y* mutants. Testis sizes and sperm counts were indistinguishable from littermate controls (Supplementary Fig. 5e–g); the frequency of spermatocyte apoptosis was not significantly increased (Supplementary Fig. 5h); the distribution of prophase I stages was not substantially altered (Supplementary Fig. 5i); and the appearance and disappearance of foci of strand exchange proteins occurred normally, with the exception that both mutants showed a statistically significant but quantitatively small reduction in the number of DMC1 foci at zygonema (Fig. 5e–h). These findings show that meiotic recombination is robust in the face of even substantial deviations from normal resection lengths (e.g., up to the ~47% reduction in *Mre11-ATLD1*).

### Side-by-side spatial organization of RAD51 and DMC1 is maintained in *Nbs1ΔB* mice

We tested whether the shortened resection lengths in *Nbs1ΔB* mice altered the occupancy and spatial disposition of ssDNA binding proteins RPA, DMC1, and RAD51 by performing ChIP followed by ssDNA sequencing (SSDS)[65,66]. In wild type, DMC1 tends to bind closer to the broken end, RAD51 binds nearer to the ssDNA/dsDNA boundary, and RPA can be found essentially anywhere along the ssDNA[65] (Fig. 6a). The SSDS signal for all three proteins was more compressed toward hotspot centers in the *Nbs1ΔB* mutant, as expected from the decreased length of ssDNA available, but the spatial pattern of DSB-proximal DMC1 and DSB-distal RAD51 was maintained (Fig. 6b).

Previous studies in yeast and mice showed that foci of RAD51 and DMC1 often appear side by side[65,67,68]. We tested using structured illumination microscopy (SIM) whether this organization at individual DSB sites was retained in *Nbs1ΔB* mice, comparing homozygous mutants with heterozygous littermate controls (which have nearly normal resection lengths (Supplementary Fig. 6a)). Similar to previous results[65], a majority of the foci of both DMC1 (93.8%) and RAD51 (79.4%) were located close to chromosome axes in *Nbs1^{ΔB/+}* heterozygotes (Figs. 6c and Supplementary Fig. 6b and Supplementary Table 1). Moreover, the foci often occurred together in pairs: 80.7% of axis-associated DMC1 foci and 77.4% of axis-associated RAD51 foci were within 320 nm of the other protein (Fig. 6c and Supplementary Figs. 6c and Supplementary Table 1). The high degrees of both axis association and colocalization were retained in *Nbs1^{ΔB/ΔB}* homozygotes (94.7% axis association for DMC1 and 86.3% for RAD51; 84.8% colocalization for DMC1 foci and 82.3% for RAD51) (Fig. 6c and Supplementary Fig. 6b, c and Supplementary Table 1).

DMC1 and RAD51 foci were nearly always offset from one another, with a distance in the controls of 103.3 ± 67.5 nm (mean ± SD) between DMC1 foci and their RAD51 mates (Fig. 6d and Supplementary Table 1). This distance was decreased significantly in *Nbs1ΔB* mutants, but with a small effect size (96.8 nm ± 66.5).

Two conclusions emerge. First, the decreased distance between DMC1 and RAD51 foci when resection lengths are reduced supports the interpretation that paired foci represent binding of both proteins to

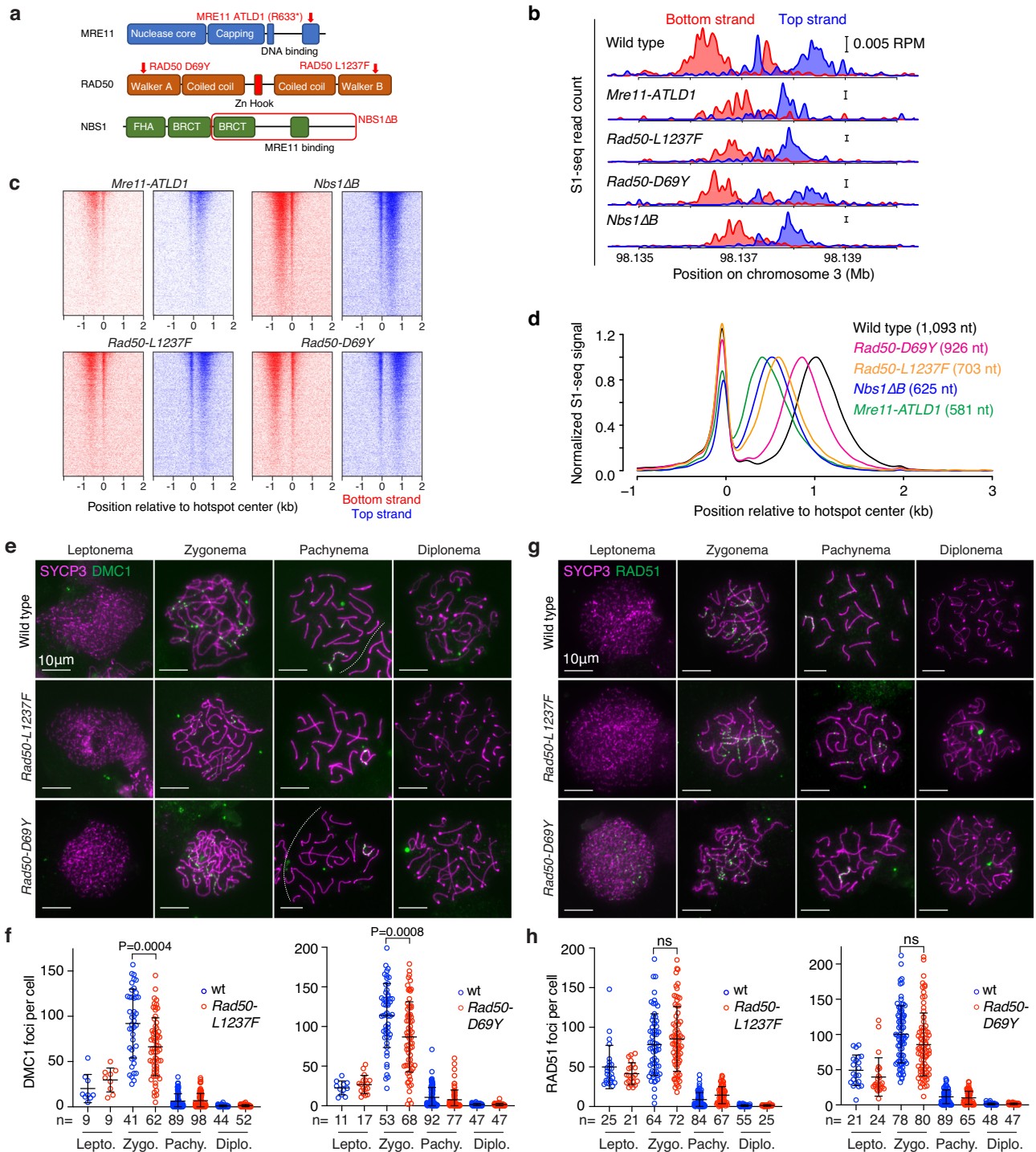

**Fig. 5 | Reduced resection length in MRN hypomorphs. a** Schematic of MRN hypomorphs. MRE11 R632* (R632 → stop) mimics human R633* identified in ATLD patients[21]. *Nbs1ΔB* produces an 80 kDa protein (red box), mimicking the *657del5* allele found in 95% of NBS patients[26]. **b** S1-seq at the hotspot from Fig. 1f, from 14.5-dpp mice. Wild type is reproduced from Fig. 1f. Two biological replicates were averaged for each of the other genotypes. Scale bars, 0.005 RPM. **c** Heatmaps (data in 40-bp bins) of S1-seq reads around DSB hotspots. **d** Genome-average S1-seq at hotspots from 14.5-dpp mice. Wild type is reproduced from Fig. 2a. Data are smoothed with a 151-bp Hann window and normalized as described in Fig. 4d. **e–h** Minimal if any changes in RAD51 and DMC1 focus numbers during meiosis in *Rad50-L1237F* or *Rad50-D69Y* mice. Representative spreads stained for SYCP3 and DMC1 (**e**) and RAD51 (**g**) along with quantification (**f, h**) are shown. Each dot in the graphs is the focus count from a single nucleus; error bars indicate means ± SD. Results of unpaired t tests (two-sided) are shown for zygonema; differences at the other stages were not statistically significant (P > 0.05). Source data are provided as a Source data file.

the same ssDNA tail and not to separate DSB ends, similar to the situation inferred from yeast and mouse studies[67,68]. Second, the effect of the *Nbs1ΔB* mutation on interfocus distances (less than 7% decrease on average) was much smaller than the effect on both resection length (43% decrease, Fig. 5d) and ChIP profiles (Fig. 6b). This difference

between cytology and population-average molecular assays shows that ssDNA length is not the most important determinant of the three-dimensional spatial organization of DSB repair foci. This conclusion in turn fits with the idea that individual DMC1 and RAD51 filaments are fairly short, estimated at ~100 nt in yeast and ~400 nt in mouse[67,68].

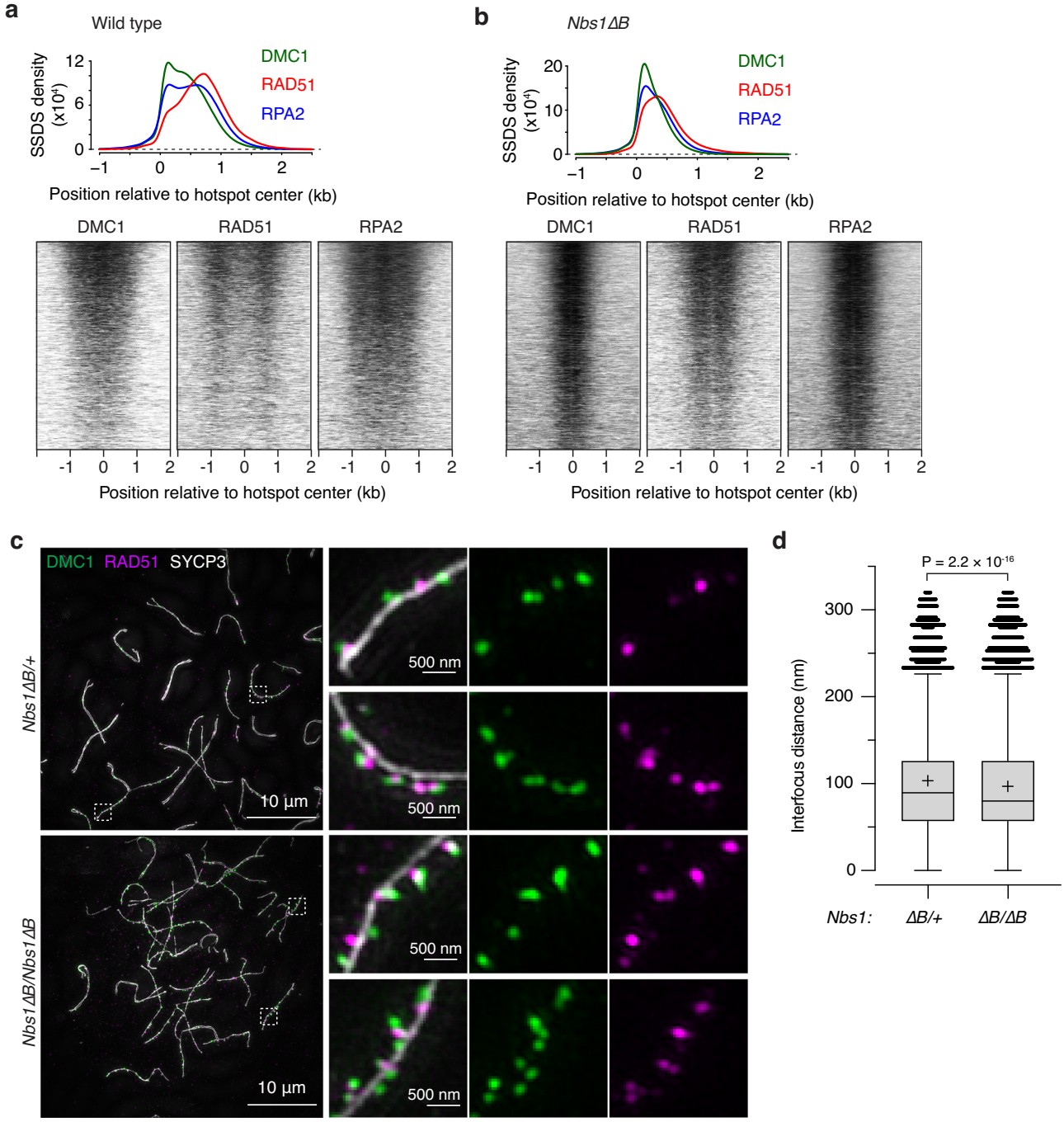

**Fig. 6 | Side-by-side spatial organization of RAD51 and DMC1 is maintained despite shortened resection in *Nbs1ΔB* mice. a, b** Genome-wide profiles of DMC1, RAD51, and RPA2 ChIP followed by SSDS in wild type (**a**) or *Nbs1ΔB* (**b**). In the graphs above, top- and bottom-strand reads around hotspots were co-oriented, combined and averaged. These maps represent sequencing coverage rather than endpoint counts reported for S1- or Exo7/T-seq. Data are smoothed with a 151-bp Hann window. An estimated background was removed by subtracting the value of signal 2.5 kb away from the hotspot center, then profiles were normalized by setting the area under each curve (from −1.0 to 2.5 kb) to 1. Heatmaps below show SSDS signals in 40-bp bins, locally normalized by dividing each signal by the total signal in a 4001-bp window around each hotspot's center. Each hotspot thus has a total value of 1, so that spatial patterns can be compared between hotspots of different strengths. **c** Representative SIM images of spread spermatocytes from 14.5-dpp mice. Insets show zoomed views of the regions indicated by dashed boxes. **d** DMC1 to RAD51 distances for axis-associated co-foci. Tukey box plots are shown for DMC1 foci that were ≤ 450 nm from an axis and ≤ 320 nm from the nearest RAD51 focus. The plus signs indicate means; horizontal lines are medians; box edges are interquartile range; whiskers indicate the most extreme data points which are ≤ 1.5 times the interquartile range from the box; individual points are outliers. P value is from a t test (two-sided). The number of images and foci included in the analysis was summarized in Supplementary Table 1. Source data are provided as a Source data file. To

Because filaments are only long enough to occupy a small fraction of the available ssDNA, even a substantial reduction in resection lengths should not cause a proportionate decrease in the size of cytologically observable foci.

## Additive contributions of EXO1 and the NBS1 N terminus to resection length

In vitro, yeast MRX promotes Exo1 activity by providing nicks for Exo1 to act on and by fostering Exo1 binding to DNA ends[69–72]. To

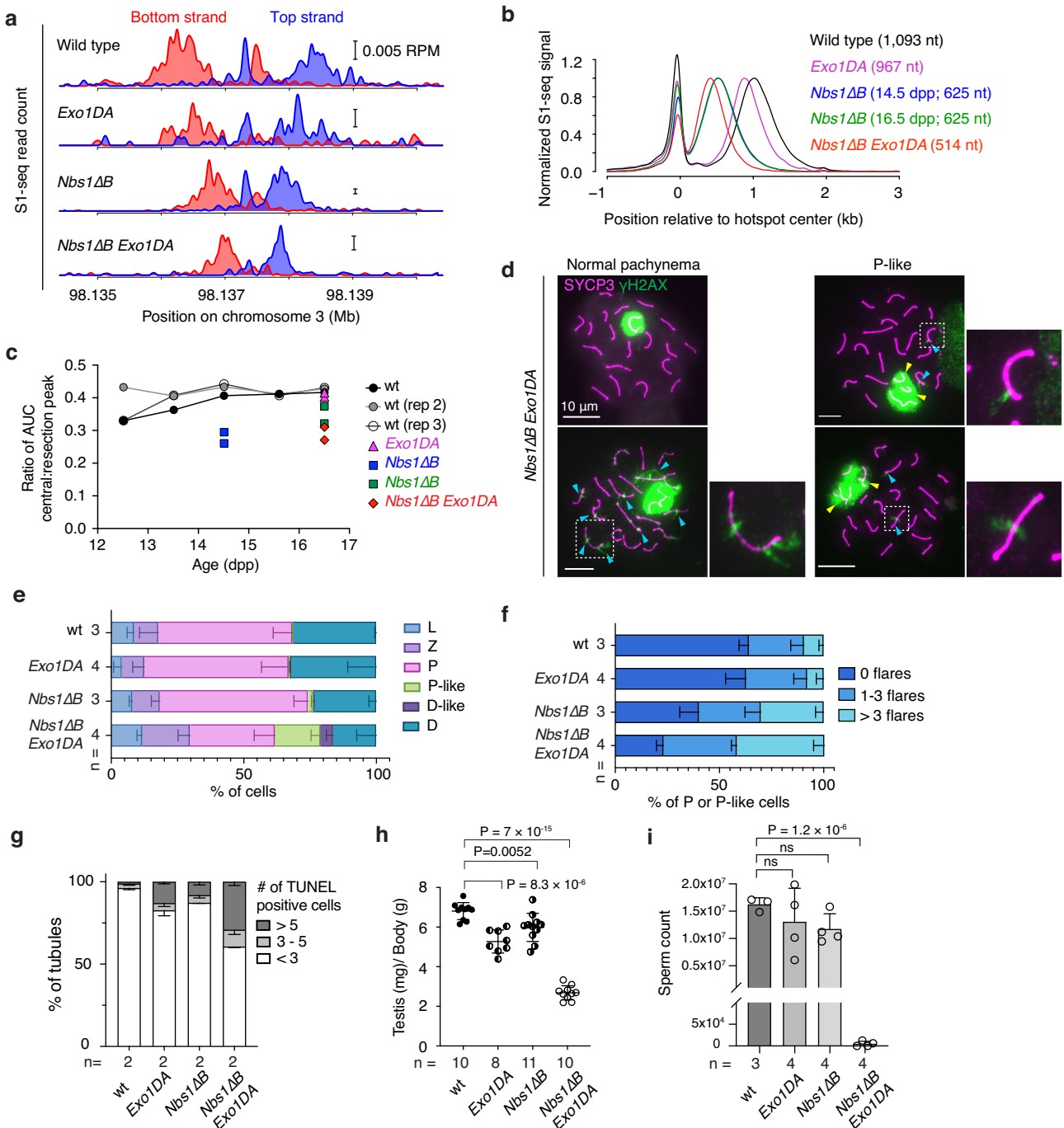

**Fig. 7 | *Nbs1ΔB* and *Exo1DA* mutations affect resection additively. a** S1-seq at the hotspot from Fig. 1f. Because of the smaller size of *Nbs1ΔB Exo1DA* pups, we collected samples at 16.5 dpp for this experiment. Wild type (14.5 dpp) is reproduced from Fig. 1f; two biological replicates were averaged for each of the other genotypes. Scale bars, 0.005 RPM. **b** Genome-average S1-seq patterns at hotspots, smoothed with a 151-bp Hann window. Wild type and *Nbs1ΔB* (14.5 dpp) are reproduced from Fig. 5d. Mean resection lengths are indicated. **c** Reduced central S1-seq signal (presumed recombination intermediates) in *Nbs1ΔB* mutants. The plot shows the relative amount of central signal calculated as the area under the curve (AUC) for the central peak (−300 to +100 bp relative to hotspot centers) divided by the AUC for the resection peak ( + 100 to +2500 bp). For wild type, in each of three independent time courses, samples were collected from a single litter daily at approximately the same time each day. **d** Homolog pairing and/or synapsis

defects. Representative spreads of *Nbs1ΔB Exo1DA* spermatocytes are shown with either normal-appearing SYCP3 and γH2AX staining or with unsynapsed or tangled chromosomes within the γH2AX-stained domain (yellow arrowheads). Blue arrowheads point to examples of γH2AX flares on synapsed autosomes.
**e** Spermatocyte stages based on SYCP3 and γH2AX staining. Error bars indicate mean ± SD of the indicated number of animals. **f** Percentage of pachytene or pachytene-like cells with no, 1–3, or > 3 γH2AX flares. Error bars indicate mean ± SD of the indicated number of animals. **g** Frequencies of tubules with the indicated number of TUNEL-positive cells. Error bars indicate mean ± range for the indicated number of animals. **h** Reduced testis size in *Nbs1ΔB Exo1DA* double mutants (5–11 wk old). Error bars indicate mean ± SD. *P* values are from Student's t tests (two-sided). **i** Sperm counts (mean ± SD). Results of Student's t tests (two-sided) are shown. Source data are provided as a Source data file.

query the interplay between MRN and EXO1 in meiotic resection, we generated double mutant mice homozygous for both *Nbs1ΔB* and *Exo1* mutations. An *Exo1* null is synthetically lethal with *Nbs1ΔB*[73], so we used the nuclease-deficient *Exo1-D173A (DA)* allele[74], reasoning that the milder somatic phenotype of *Exo1DA*[75] might allow us to obtain double mutants. Indeed, *Nbs1ΔB Exo1DA* doubly homozygous mice were viable, albeit recovered at a sub-mendelian frequency (Supplementary Fig. 7a) and exhibiting lower body weights (Supplementary Fig. 7b).

*Nbs1ΔB Exo1DA* double mutants showed a 111-nt decrease in the average resection length compared to *Nbs1ΔB* alone (Figs. 7a, b and Supplementary Fig. 7c). Since this is similar to the 126-nt average decrease in *Exo1DA* compared to wild type, EXO1 appears to work additively with MRN. We also conclude that the *Nbs1ΔB* resection defect is not attributable to a defect in EXO1 activity.

Relative to the S1-seq signal from resection endpoints, the central signal from recombination intermediates appeared lower in *Nbs1ΔB Exo1DA* than in wild type or the single mutants (Fig. 7b). To evaluate this, we measured total read count (area under the curve) separately for the central peak and for the resection endpoint peak and calculated the ratio between these values (Fig. 7c). In wild type, the central peak was reproducibly ~40% of the level of the resected DSBs from 13.5 to 16.5 dpp. This ratio was lower at 12.5 dpp (~33%) in two of three experiments (Fig. 7c). Since this earlier age is likely to be more enriched for earlier prophase I spermatocytes, the lower relative amount of central signal may reflect timing differences between DSB formation (appearance of the resected DSB signal) and formation of the recombination intermediates detected by S1-seq.

In *Nbs1ΔB* mice, the relative amount of central signal was substantially lower at 14.5 dpp (27.5% of the DSB signal; mean of $n = 2$) and it increased but remained lower than wild type at 16.5 dpp (34.7%; Fig. 7c). At 14.5 dpp, *Mre11-ATLD1* exhibited a reduction similar to *Nbs1ΔB*, but *Rad50-L1237F* and *-D69Y* did not (Supplementary Fig. 7d). The central signal at 16.5 dpp was even lower in *Nbs1ΔB Exo1DA* (28.8%) but the *Exo1DA* mutation by itself had no effect. These decreased yields of the central signal in *Mre11-ATLD1*, *Nbs1ΔB*, and *Nbs1ΔB Exo1DA* may indicate that fewer of the relevant recombination intermediates are formed, that they turn over more rapidly, and/or that they exhibit structural features that make them more difficult to detect by S1-seq. Regardless of the source of the changes, however, it appears that *Mre11-ATLD1* and *Nbs1ΔB* mutants experience alterations in interhomolog recombination, with at least the *Nbs1ΔB* mutant exacerbated when combined with *Exo1DA*.

Reinforcing this view, *Nbs1ΔB Exo1DA* double mutants displayed an elevated frequency of pachytene-like cells with synaptic failures (tangled and/or unpaired chromosomes) and a higher proportion of zygotene cells than normal, possibly indicative of delays in completing synapsis (Fig. 7d, e). Pachytene and pachytene-like cells from the double mutant also showed an increased frequency of γH2AX flares on synapsed chromosomes, indicative of delays in DSB repair (Fig. 7d, f). Seminiferous tubule sections showed few if any elongating spermatids in *Nbs1ΔB Exo1DA* double mutants (Supplementary Fig. 7e) and an elevated frequency of apoptotic spermatocytes (Fig. 7g and Supplementary Fig. 7f). As a consequence of this germ cell loss, the mice had substantially reduced testis sizes (Fig. 7h) and were essentially devoid of epididymal sperm (Fig. 7i and Supplementary Fig. 7e). Single *Nbs1ΔB* or *Exo1DA* mutants showed few of these defects, consistent with previous studies[60,75], although apoptotic tubules were observed at modestly elevated frequencies and testis sizes were reduced (Fig. 7g, h).

## Discussion

Unlike the paradigmatic budding yeast pathway for meiotic DSB resection[4,12], mice rely less on EXO1[14,15] and we now show that multiple MRN mutations confer drastic resection defects (see also[20]). Thus, while the canonical two-step handoff from MRN(X) to EXO1 occurs in both yeast and mammals, we find that mouse MRN is essential not just

for resection initiation, but also unexpectedly for most of the normal length of resection as well. Since *Exo1DA* combined additively with MRN deficiency, EXO1 appears to "polish" resection tracts that are generated primarily through MRN.

ATM deficiency decreases resection lengths for a subset of DSBs[14,15]. Thus, because the MRN hypomorphs compromise ATM activation[21,26,76], their resection defects could be an indirect consequence of diminished ATM signaling. However, *Mre11-cHN* mice appear to have normal ATM activity[29,54,57,58] (Fig. 4f), suggesting that their defective resection is tied more directly to attenuated MRN activity per se. By extension, the resection defects in MRN hypomorphic adults and *Mre11-cKO* juveniles may be explained in a similar manner. We cannot exclude that reduced ATM activity also contributes, but *Mre11-ATLD1* and *Nbs1ΔB* mutants only modestly attenuate meiotic ATM activation as judged by their retention of most ATM-dependent DSB control[77], so this explanation would require that resection is particularly sensitive to small decreases in ATM activation. Moreover, the resection-promoting function of ATM itself remains unknown, so the opposite causality is also plausible: i.e., resection defects in *Atm* mutants might hypothetically be an indirect consequence of losing ATM-dependent regulation of MRN.

We consider two plausible ways in which MRN might promote the extension of resection tracts aside from ATM activation. First, it might directly resect DNA via iterative nicking that spreads progressively from the DSB end. Although yeast MRX and human MRN preferentially cleave the 5′-terminal DNA strand ~15–45 nt away from a blocked DNA end in vitro[78–81], MRN can both oligomerize and diffuse on DNA[82,83]. Moreover, MRX can resect unrepairable DSBs in vivo more than one kilobase in yeast lacking long-range resection enzymes[84], and iterative nicking was proposed to explain both short-range resection in vegetative yeast cells[85] and the ~300 nt average resection in meiotic cells lacking Exo1 activity[4]. If this interpretation is correct, our results indicate that the iterative action of MRN can extend resection up to ~1500 nt in mice when the nuclease activity of EXO1 is compromised.

A second, non-exclusive possibility is that MRN recruits or activates another nuclease(s). MRN enhances DNA2 nuclease in vitro[69,86–89], although it remains unclear if the same is true in vivo. CtIP also promotes DNA2 activity both in vitro and in vivo[90,91]. Budding yeast Dna2 is unlikely to be important for meiotic resection because its partner helicase Sgs1 is dispensable[4,11,12]. However, the possibly mammal-specific interaction between CtIP and DNA2[90,91] may make DNA2 a candidate for long-range resection in mouse meiosis[15].

MRN involvement in resection initiation was not surprising. However, the specific genetic dependencies are intriguing. For example, we were surprised to observe only minor apparent defects in resection initiation in *Mre11-cHN* mice. It is formally possible that MRE11 nuclease is dispensable for clipping SPO11 from DSBs even though MRE11 protein itself is required. However, we consider it more likely that this result instead reflects continuing presence of sufficient wild-type MRE11. For example, dimerization with continuously expressed MRE11-H129N might slow the degradation of wild-type protein compared to what happens in *Mre11-cKO*. In this scenario, mutant/wild-type heterodimers are still active for strand cleavage since they retain a functional nuclease active site. Alternatively, MRE11-H129N might amplify activity of a small amount of residual wild-type protein in a structural role via heterodimer formation or oligomerization on DNA. It is also possible that binding of inactive MRN complexes to DSBs allows another nuclease(s) to be recruited to initiate resection in an unconventional manner.

Interestingly, the meiotic progression defects of the *Mre11-cHN* mutant were considerably worse than those in *Mre11-ATLD1* and *Nbs1ΔB* despite *Mre11-cHN* having a longer average resection tract length. It is possible that the meiotic progression defects in *Mre11-cHN* reflect an as yet unknown function for MRE11 nuclease activity in downstream steps of recombination. However, it is also possible that

the progression defects are entirely consequences of defects in resection initiation and/or DSB formation. For example, the reduced SPO11-oligo complexes and hotspot-associated sequencing signal may indicate that there are fewer processed DSBs available to support recombination and chromosome pairing. If so, such a reduction in processed DSBs would also explain the apparent delay during leptonema and zygonema and sporadic synapsis failure (just a few chromosomes per cell). Indeed, we previously showed that reducing DSBs to about half the normal level gives a meiotic progression defect that is quite similar to the one we document here for *Mre11-cHN*[92].

The *Mre11-cKO* mutant is the first example in mammals of a yeast *rad50S*-like meiotic phenotype, namely, accumulation of completely unresected DSBs[7,93–95]. This finding definitively establishes that MRN is essential for resection initiation. The elevated DSB sequencing signal and double cutting in this mutant also confirms and extends the previous conclusion, based on modest elevation of SPO11-oligo complexes in *Mre11-ATLD1 and Nbs1ΔB*, that MRN promotes ATM activation in the context of controlling SPO11 activity[77]. While this manuscript was in preparation, conditional deletion of *Rad50* (with *Stra8-Cre*) was also found to cause an increased frequency of unresected DSBs, but to a much more modest extent than we observed with *Mre11-cKO* and without an apparent increase in DSB levels or evidence of double-cutting[20]. Because resection was not directly assessed for *Nbs1* conditional deletion (with *Vasa-Cre*), it is unclear whether NBS1 is required for resection initiation[19]. The differences between the conditional mutants may reflect differences in timing and efficiency of gene deletion and protein disappearance rather than different requirements for the individual MRN subunits.

Our findings may also indicate that mouse MRE11 is dispensable for DSB formation, unlike in *S. cerevisiae* and *Caenorhabditis elegans*, but similar to *Schizosaccharomyces pombe*, *Drosophila melanogaster*, and *Arabidopsis thaliana* [reviewed in ref. [96]]. However, this interpretation should be viewed cautiously because *Mre11-cKO* spermatocytes may retain trace amounts of MRE11 protein that are sufficient for DSB formation but not resection. In this vein, we note that *Mre11* and *Ctip* null mutations are both cell-lethal, whereas conditional deletion of *Mre11* but not of *Ctip* was tolerated by spermatogonia and premeiotic spermatocytes. This is consistent with both proteins being essential for mitotic germ cell divisions, but MRE11 protein persisting longer than CtIP after Ngn3-Cre-mediated excision.

In addition to defining the genetic control of DSB resection, our findings offer insight into the minimal resection length needed for successful downstream steps. The extensive pairing and synapsis defects in *Mre11-cKO* mice met the expectation that recombination cannot proceed without ssDNA generated by resection. However, our findings showed that meiosis is remarkably robust as long as some resection occurs. Recombination, chromosome pairing, synapsis, and chromosome segregation all occurred efficiently with slightly more than half the normal average resection length (*Mre11-ATLD1*). Severe problems were seen only in the *Nbs1ΔB Exo1DA* double mutant, with its only modest further decrement in resection length. But even in this mutant, although most cells failed to complete meiosis, most chromosomes paired and synapsed well, indicating that most DSBs were successfully engaging in recombination.

It is unclear why recombination begins to fail in the *Nbs1ΔB Exo1DA* double mutant but not appreciably in the *Nbs1ΔB* or *Mre11-ATLD1* single mutants. The minimal length of ssDNA needed to support homology searching and strand exchange in mammalian meiosis is unknown, but vegetative yeast cells can execute recombination with less than 40 bp of homology, and normal recombination efficiency appears to need only ~100–250 bp[97–101]. Moreover, RAD51 and DMC1 filaments are thought to occupy only a few hundred nucleotides[67,68]. Although the distribution of resection tract lengths is highly stereotyped across all mouse hotspots, there is substantial variation from

DSB to DSB, and perhaps between the two sides of a single DSB. It has been suggested for yeast that normal resection lengths ensure that all DSBs are resected enough to support efficient, accurate recombination[12]. It is therefore possible that *Nbs1ΔB Exo1DA* double mutants become vulnerable to recombination defects because a large enough subset of their DSBs fall below the minimum length needed for strand exchange per se or for proper choice of homologous recombination partner[102].

## Methods

### Mice
Experiments conformed to the US Office of Laboratory Animal Welfare regulatory standards and were approved by the Memorial Sloan Kettering Cancer Center Institutional Animal Care and Use Committee (Protocol number 01-03-007). Mice were housed in solid-bottom, polysulfone, individually ventilated cages (IVCs) (Thoren Caging Systems, Hazelton, PA) on autoclaved aspen-chip bedding (PWI Industries Canada, Quebec, Canada); γ-irradiated feed (LabDiet 50531, PMI, St Louis, MO) and acidified reverse osmosis water (pH 2.5 to 2.8) provided ad libitum. The cages also contained Nestlets®, EnviroDri®, and/or EnviroPaks® as environmental enrichment. The IVC system was ventilated at approximately 30 air changes hourly. HEPA-filtered room air was supplied to each cage and the rack effluent was exhausted directly into the building's exhaust system. Cages were changed weekly in either a HEPA-filtered vertical flow change station or a Class 2 Type A biological safety cabinet. The animal holding room was maintained at $21.5 \pm 1\,°C$, relative humidity between 30% and 70%, and a 12:12 hour light:dark photoperiod. Mice were euthanized by $CO_2$ asphyxiation prior to tissue harvest. Previously described *Mre11-flox* and *Mre11-H129N*[29] and *Ctip-flox*[103,104] alleles were crossed with *Ngn3-Cre*[30] to create male germline-specific mutations. No randomization or blinding was performed.

To generate *Mre11-cKO* experimental mice, *Mre11^{flox/flox}* (or *Mre11^{wt/flox}* based on availability) mice were crossed with *Mre11^{+/−} Ngn3-Cre^{+}* mice. The *Mre11-deletion* allele (*Mre11^{−}*) was derived by Cre-mediated recombination of the *Mre11-flox* allele. To generate *Mre11-cHN* experimental mice, *Mre11^{H129N/+} Ngn3-Cre^{+}* mice were crossed with *Mre11^{flox/flox}* mice. To generate *Ctip-cKO* mice, *Ctip^{flox/flox}* mice were crossed with *Ctip^{+/−} Ngn3-Cre^{+}* mice. The *Ctip* deletion allele (*Ctip^{−}*) was derived by Cre-mediated recombination of *Ctip-flox*.

Although *Ngn3-Cre* is also expressed in embryonic pancreas and endocrine cells[30], we did not observe any gross defects of mutant animals created with *Ngn3-Cre* up through the ages of testis sample collection. Previously described *Atm*[105] *Spo11*[106], *Rad50S*[23], *Mre11-ATLD1*[21], *Nbs1ΔB*[26], and *Exo1DA*[74] mutations were maintained on a congenic C57BL/6 J strain background. *Nbs1ΔB Exo1DA* animals were generated by crossing *Nbs1ΔB* mice with *Exo1DA* mice. To enhance the survival rate of double homozygous male mice for experimental purposes, female pups were euthanized by hypothermia between 4–6 dpp in accordance with the approved animal protocol.

To generate targeted *Rad50* mutations, a guide RNA cassette with sequence (5′-TACATTTGTTCATGATCCCA(AGG)) and single-stranded donor DNA (5′- TTATAAAAATTTAATTCTTAAAATACAAAACTTTACAG ACACCATTACCTTGGGATaATGAACAAATGTATTTCCTTTGGTTCCAG GAGGGAAATCTCCAGTACAA) harboring desired missense mutations for D69 > Y, or a guide RNA cassette with sequence (5′-TCTCGGTCG AGATTTGTTGT(CGG)) and single-stranded donor DNA (5′- GGAAACC TTCTGTCTGAACTGTGGCATCCTTGCCTTGGATGAGCCGACAACAAA TtttGACCGAGAAAACATTGAGTCTCTTGCACATGCTTTGGTTGAGTAA GTA) harboring desired missense mutations for L1237 > F were microinjected into pronuclei of zygotes. Founder mice were crossed to C57BL/6 J mice purchased from Jackson Laboratories to obtain germline transmission, then heterozygous animals were backcrossed to C57BL/6 J for at least three generations.

## S1-seq and Exo7/T-seq

**Choice of sequencing method and animal ages.** S1-seq was originally applied to juvenile testis samples because these samples lack post-meiotic cells and are thus enriched for spermatocytes that contain DSBs and recombination intermediates[15]. We continued to use juvenile samples for most experiments here, but found that Cre-mediated excision of the *Mre11-flox* construct was more penetrant in adult samples. Initial attempts to perform S1-seq on adult samples gave poor signal strength from SPO11-generated DSBs at hotspots because of the presence of S1-sensitive triplex secondary structures in the DNA[107]. We therefore initially used Exo7/T-seq for analysis of resection in *Mre11-cKO* and *Mre11-cHN* mice, because exonuclease VII and exonuclease T require DNA ends and thus do not give sequencing reads from many of the non-DSB structures that S1-seq can pick up. We later found that increasing the pH during the nuclease S1 digestion step greatly improved the specificity for SPO11-initiated events by avoiding the formation of triplex structures[34], so we repeated the analysis of *Mre11-cKO* using S1-seq on adult samples. For the wild-type time courses (Fig. 7c), each time point was processed up to the agarose plug step, then S1-seq maps were prepared in a single batch after collecting the last time point.

**Library preparation and sequencing.** Libraries were prepared as described elsewhere[34], based on S1-seq[15] and END-seq[14,108]. Briefly, testes from mice at the indicated ages were decapsulated and digested with collagenase type IV (Worthington) and Dispase II (Sigma). The resulting seminiferous tubule preparations were then further treated with TrypLE Express enzyme (GIBCO) and DNase I (Sigma) for 15 min at 35 °C. TrypLE enzyme was inactivated with 5% FBS and tubules were further dissociated by repeated pipetting. Cells were then passed through a 70-μm cell strainer (BD Falcon).

One to two million cells from juvenile testes or two to three million cells from adult testes were embedded in plugs of 1% low-melting-point agarose (Lonza). After a brief incubation at 4 °C until the agarose became solid, plugs were incubated with proteinase K (Roche) in lysis buffer (0.5 M EDTA at pH 8.0, 1% N-lauroylsarcosine sodium salt) at 50 °C over two nights. Plugs were washed five times for 20 min with TE (10 mM Tris-HCl at pH 7.5, 1 mM EDTA at pH 8.0), and then incubated with 100 μg/ml RNase A (Thermo) for 3 hr at 37 °C. Plugs were then washed five times with TE and stored in TE at 4 °C until usage.

All library types were prepared in the same way aside from the ssDNA digestion step. For S1-seq, plugs were equilibrated with S1 buffer (50 mM sodium acetate, 280 mM NaCl, 4.5 mM ZnSO$_4$, pH 4.7 at 25 °C), then treated with 9 U of nuclease S1 (Promega) for 20 min at 37 °C. For Exo7/T-seq, plugs were instead equilibrated with exonuclease VII buffer (50 mM Tris-HCl, 50 mM sodium phosphate, 8 mM EDTA, 10 mM 2-mercaptoethanol, pH 8 at 25 °C), then treated with 50 U of exonuclease VII (NEB) for 60 min at 37 °C. Plugs were then equilibrated in NEBuffer 4 and treated with 75 U of exonuclease T (NEB) for 90 min at 24 °C.

Plugs were next equilibrated with T4 polymerase buffer (T4 ligase buffer (NEB) supplemented with 100 μg/ml BSA and 100 μM dNTPs (Roche)) then incubated with 30 U T4 DNA polymerase (NEB) at 12 °C for 30 min. Plugs were then washed in TE and equilibrated in T4 ligase buffer on ice. Biotinylated P5 adapters were ligated to the blunted ends with 2000 U T4 DNA Ligase (NEB) at 16 °C for 20 hr. After ligation, plugs were soaked in TE overnight at 4 °C to diffuse excess unligated adapters out of the plugs. The plugs were then equilibrated in 1× β-agarase I buffer (10 mM Bis-Tris-HCl, 1 mM EDTA, pH 6.5), incubated at 70 °C for 15 min to melt the agarose with brief vortexing every 5 min, mixed well, cooled to 42 °C, and digested with 2 μl of β-agarase I (NEB) at 42 °C for 90 min. DNA was sheared to fragment sizes ranging 200–500 bp with a Covaris system (E220 Focused-ultrasonicator, microtube-500) using the following parameters: delay 300 s then three cycles of [peak power 175 W, duty factor 20%, cycles/burst 200,

duration 30 s, and delay 90 s]. The DNA was precipitated with ethanol, then dissolved in TE. Unligated adapters were removed with SPRIselect beads (Beckman Colter). Fragments containing the biotinylated adapter were further purified with Dynabeads M-280 streptavidin (Thermo) and DNA ends were repaired using the End-it DNA End-repair kit (Lucigen). P7 adapters were ligated to DNA fragments, then PCR was done directly on the bead-immobilized fragments. PCR products were purified with AMPure XP beads (Beckman Colter) to remove primer dimers and unligated adapters.

After Qubit (Thermo) quantification and quality control by Agilent BioAnalyzer, libraries were pooled and sequenced on the Illumina NextSeq, HiSeq, or NovaSeq platforms equipped with HiSeq Control Software (version 2.2.68) or Real Time Analysis (version 3.1 or 4.1) in the Integrated Genomics Operation (IGO) at Memorial Sloan Kettering Cancer Center. A spike-in of bacteriophage ΦX174 DNA was added to increase diversity when necessary and for quality control purposes. We obtained paired-end reads of 50 bp.

**Preprocessing and mapping.** Base calls were performed using bcl-convert software v.3.9.3 to 3.10.5 or bcl2fastq software v.2.20. Reads were trimmed and filtered by Trim Galore version 0.6.6 with the arguments --paired --length 15 (http://www.bioinformatics.babraham.ac.uk/projects/trim_galore/). Sequence reads were mapped onto the mouse reference genome (mm10) by bowtie2 version 2.3.5.1[109] with the arguments -N 1 -X 1000. Duplicated reads were removed by Picard (https://broadinstitute.github.io/picard/). Uniquely and properly mapped reads (MAPQ ≥ 20) were extracted by samtools with the argument -q 20 (http://www.htslib.org/). Reads were assigned to the nucleotide immediately next to the biotinylated adapter. Mapping statistics are in Supplementary Data 1.

**Bioinformatic analysis and plotting.** Maps were analyzed using R version 4.0.3 to 4.3.2 (http://www.r-project.org) and R studio (version2023.06.1 + 524). To generate average profiles around individual hotspots (e.g., Fig. 1f), RPM values from biological replicates were averaged before smoothing and plotting. For genome-average profiles around hotspots (e.g., Fig. 2a), bottom-strand reads (in RPM) were co-oriented with top-strand reads by flipping the coordinates relative to the hotspot midpoint, then the combined top- and bottom-strand reads were averaged across all hotspots before smoothing and plotting. For genotypes in which resection was initiated (i.e., all genotypes except *Mre11-cKO*), we further normalized the genome average profiles as follows. First, an estimated background was removed by subtracting from all values the value of signal 2500 bp away from the hotspot center. The signal was then normalized to the peak height of resection endpoints, i.e, the maximum value among positions from 100 to 2,500 bp. Negative values were set as zero for plotting purposes. Where appropriate, data were smoothed with a Hann function.

To calculate the mean resection length, S1-seq signal was averaged across hotspots and an estimated background was removed by subtracting from all values the value of signal 2.5 kb away from the hot spot center. The signal close to and further away from the hot spot center was excluded by setting values of positions <100 bp and >2.5 kb to zero. Fractions of total signal were calculated every 100 bp and the mean resection length was calculated. The modal resection length in Supplementary Fig. 2f was determined by the peak position of the genome-average profile between 100 bp and 2500 bp from hotspot centers.

The color scale of heatmaps were defined after local normalization of sequencing signal at each hotspot. Signals in 40-bp bins were divided by the total signal in a 4,001-bp window around each hotspot center, so that each row has a total value of 1 regardless of the strengths of hotspots and color reflects the local spatial pattern. Normalized signals were classified into 10 groups by deciles and color-coded.

We provide analysis of variation between biological replicate maps in a separate study[34]. As shown there, the spatial patterns of resection and the central signal as well as the ratio of central signal to resection signal are highly reproducible between replicates (coefficient of variation of <1.4% for average resection length and < 6.8% for central:resection ratio from n = 10 wild-type S1-seq replicates). This reproducibility arises in part because these are internally controlled comparisons within a genomic map, so they are well-suited for RPM-based analysis. The absolute RPM values around hotspots are also reproducible, but somewhat less so (coefficient of variation of 26.9%) because of differences in signal:noise ratio between sequencing libraries (i.e., differences in the amount of SPO11-independent background). Because of the latter variability, we often ignore the absolute value of the RPM map, and instead focus on spatial patterns. Exceptions are cases where the difference in absolute RPM exceeds the normal range of sample-to-sample variation, and where we have clear evidence of reproducibility of differences. For example, the *Mre11-cKO* mutant has such a large increase in the amount of signal within hotspots that we can confidently make conclusions based on the comparison of total RPM values with wild type.

### MRE11 ChIP-seq
MRE11 ChIP was performed following the previously published DISCOVER-seq protocol[110]. In brief, fresh testes of 5–7 wk old wild-type, *Atm*−/−, or *Mre11-cHN* mice were dissociated into single cells and resuspended in DMEM. PFA (16% stock, Pierce) was added to a final concentration of 1% and incubated for 15 min with gentle agitation, then the crosslinking was stopped by adding 2.5 M glycine to a final concentration of 125 mM, followed by 3 min incubation on ice. The nuclei were prepared as described[110] and up to 10 million nuclei were resuspended in 500 µl lysis buffer (0.5% *N*-lauroylsarcosine, 0.1% sodium deoxycholate, 10 mM Tris-HCl pH 8.0, 100 mM NaCl, 1 mM EDTA) supplemented with protease inhibitor (Roche), and sonicated to fragment sizes ranging 200–300 bp with a Covaris system (E220 Focused-ultrasonicator, microtube-500) using the following parameters: delay 300 s then six cycles of [peak power 75 W, duty factor 10%, cycles/burst 200, duration 150 s, and delay 60 s]. Sheared nuclei were centrifuged at 20,000 *g* for 10 min at 4 °C to remove debris and Triton-X (1% (vol/vol) final concentration) and lysis buffer were added to make the total volume 3 ml.

For each immunoprecipitation, 10 µl of anti-MRE11 (Novus Biologicals NB100-142) was pre-mixed with 100 µl protein A Dynabeads (Thermo), then added to chromatin lysates and incubated overnight at 4 °C with end-over-end rotation. Immunoprecipitated DNA was eluted and crosslinks were reversed as described[110]. ChIP sequencing libraries were prepared from cleaned DNA as described for SSDS libraries[111] without the boiling step after the end-repair and dA-tailing steps. After ligation with sequencing adapters, the library was amplified by 15 cycles of PCR. The amplified library was purified and sequenced on the Illumina NovaSeq platform equipped with Real Time Analysis (version 3.1 or 4.1) in the IGO. We obtained paired-end reads of 100 bp.

Base calls were performed using bcl-convert software v.3.9.3 to 3.10.5 or bcl2fastq software v.2.20. Sequencing reads were processed and mapped as described for resection libraries and uniquely and properly mapped reads were counted, and maps were further analyzed using R version 4.0.3 to 4.3.2 (http://www.r-project.org) and R studio (version2023.06.1 + 524). Mapping statistics are in Supplementary Data 1.

### Single-stranded DNA sequencing (SSDS) library preparation and data analysis
ChIP followed by SSDS was performed as previously described[65,111]. In brief, a decapsulated, fresh or fresh-frozen testis of 8-12 wk old wild-type or *Nbs1ΔB* mice was placed in room temperature 1% PFA (16% stock, Pierce) in PBS and incubated for 15 min with gentle agitation.

The crosslinking was stopped by adding 2.5 M glycine to a final concentration of 125 mM, followed by 5 min incubation at room temperature. Fixed cells were immediately homogenized in a Dounce homogenizer with 20 strokes of the tight-fitting pestle, then passed through a 70 µm cell strainer. The nuclei were prepared as described[111], resuspended in 900 µl shearing buffer (0.1% SDS, 10 mM Tris-HCl pH 8.0, 1 mM EDTA) supplemented with protease inhibitor (Roche), split into two ~500 µl aliquots and sonicated to fragment sizes ranging 500–1000 bp with a Covaris system (E220 Focused-ultrasonicator, microtube-500) using the following parameters: delay 300 s then three cycles of [peak power 75 W, duty factor 10%, cycles/burst 200, duration 150 s, and delay 60 s]. Sheared nuclei were centrifuged at 12,000 *g* for 10 min at 4 °C to remove debris. The sonicated chromatin was then transferred to a 3 ml Slide-A-Lyzer G2 dialysis cassette (Pierce) and dialyzed with ChIP buffer (16.7 mM Tris-HCl, pH 8.0, 167 mM NaCl, 1.2 mM EDTA, 0.01% SDS, 1.1% Triton X-100) for 5 hr at 4 °C with constant but slow stirring. For each immunoprecipitation, 20 µl of anti-DMC1 (Abcam, ab11054), 10 µl of anti-RAD51 (Novus Biologicals, NB100-148), or 10 µl of anti-RPA2 (Abcam, ab76420) was added to chromatin lysates and incubated overnight at 4 °C with end-over-end rotation.

As described previously[111], protein G or A Dynabeads (Thermo) were added to the chromatin and antibody mixture and incubated 2 hr at 4 °C, then immunoprecipitated DNA was washed and eluted. To reverse DNA-protein crosslinks, 12 µl of 5 M NaCl was added to 300 µl of eluted sample and incubated overnight at 65 °C. DNA was cleaned with MinElute PCR Cleanup Kit (Qiagen) after proteinase K (Thermo) treatment. Following end repair and dA-tailing, DNA was incubated at 95 °C for 3 min and cooled to room temperature to enrich for single-stranded DNA. After ligation with sequencing adapters, SSDS library was amplified with 12 cycles of PCR. The amplified library was purified and sequenced on the Illumina NovaSeq platform equipped with Real Time Analysis (version 3.1 or 4.1) in the IGO. We obtained paired-end reads of 100 bp.

Base calls were performed using bcl-convert software v.3.9.3 to 3.10.5 or bcl2fastq software v.2.20. We then ran a previously described bioinformatic pipeline on the resulting reads for identification of single-stranded sequences[111]. Sequence reads at SPO11-dependent hotspots were analyzed using R version 4.0.3 to 4.3.2 (http://www.r-project.org) and R studio (version2023.06.1 + 524).

### Sperm counts
Sperm isolation was performed as described[112]. Briefly, both caudal and caput epididymis were dissected from > 5 wk old mice, with all excess fat and tissue trimmed away, and placed on top of filter mesh of 5 ml tubes with cell-strainer caps (BD) filled completely with PBS. After 5 min, the cap was carefully lifted out of the PBS to break the fluid surface tension, releasing plumes of sperm into the tube. This was repeated until sperm were no longer released after lifting out the cap. The cap (and tissue) was removed, and the tube sealed with parafilm. The sperm were pelleted in a swinging bucket rotor for 2 min at 4,000 *g*. The supernatant was aspirated down to ~1 ml, and then gently resuspended by quick vortex pulses and incubated for 1 min at 60 °C. Inactivated sperm were mixed with Trypan Bule solution and placed on a hematocytometer for counting.

### Spermatocyte chromosome spreads and immunostaining
Spreads were prepared as described with minor modifications[113]. When preparing a sequencing library and spreads from testes collected at the same time, testis cells were dissociated following the procedures in library preparation until passage through the 70 µm cell strainer (see above), then continued as described below. Briefly, decapsulated testis samples were placed in 2 ml TIM buffer (104 mM NaCl, 45 mM KCl, 1.2 mM MgSO₄, 0.6 mM KH₂PO₄, 0.1% glucose, 6 mM sodium lactate, 1 mM sodium pyruvate, pH 7.3), with 200 µl collagenase (20 mg/ml),

then shaken at 350 rpm for 15 min at 32 °C. After washing, tubules were resuspended in 2 ml TIM buffer with 200 μl trypsin solution (7 mg/ml) and 20 μl DNase I (400 μg/ml) and shaken for 15 min at 350 rpm at 32 °C. To stop trypsin digestion, 500 μl of FBS was added. Resuspended cells were filtered through a 70 μm cell strainer and TIM buffer was added to 15 ml. After three washes with DNase I, cells were resuspended in 10 ml of TIM buffer and distributed in 1 ml aliquots to Eppendorf tubes, centrifuged, and the supernatant was removed. The cell pellet was resuspended in 80 μl pre-warmed 0.1 M sucrose and incubated for 10 min. From the cell suspension, 40 μl was added to a Superfrost glass slide covered with 130 μl 1% PFA (with 0.1% Triton X-100, pH 9.2) in a humid chamber. Slides were kept in the closed humid chamber for at least 2 hr at 4 °C, then air dried and rinsed with 0.4% Photo-Flo (Kodak). After air drying, slides were wrapped in aluminum foil and stored at -80 °C.

For immunofluorescence, slides were blocked for 10 min with dilution buffer (0.2% BSA, 0.2% fish skin gelatin, 0.05% Tween-20 in 1× PBS) at room temperature. Using a humid chamber, slides were incubated with 100 μl primary antibody solution overnight at 4 °C. After washing, secondary antibody was added for 1 hr at 37 °C. After washing, slides were mounted with Vectashield containing DAPI (H-1000, Vector Laboratories). Primary and secondary antibodies are listed in Supplementary Data 2.

## Histology

Testes and epididymides dissected from adult mice were fixed in Bouin's fixative for 4 to 5 hr at room temperature, or in 4% PFA overnight at 4 °C. Bouin's fixed testes were washed in 15 ml milli-Q water on a horizontal shaker for 1 hr at room temperature, followed by five 1-hr washes in 15 ml of 70% ethanol on a roller at 4 °C. PFA-fixed tissues were washed four times for 5 min in 15 ml milli-Q water at room temperature. Fixed tissues were stored in 70% ethanol before embedding in paraffin and sectioning at 5 μm. The tissue sections were deparaffinized with EZPrep buffer (Ventana Medical Systems). Antigen retrieval was performed with CC1 buffer (Ventana Medical Systems). Sections were blocked for 30 min with Background Buster solution (Innovex), and then avidin-biotin blocked for 8 min (Ventana Medical Systems). Hematoxylin and eosin (H&E) staining and immunohistochemical TUNEL assays were performed by the MSK Molecular Cytology Core Facility using the Autostainer XL (Leica Microsystems, Wetzlar, Germany) automated stainer for H&E or PAS with hematoxylin counterstain, and using the Discovery XT processor (Ventana Medical Systems, Oro Valley, Arizona) for TUNEL. Testis sections were incubated with anti-DDX4 or anti-MRE11 for 5 hr, followed by 60 min incubation with biotinylated goat anti-rabbit (Vector Labs) at 1:200 dilution (Supplementary Data 2). Detection was performed with DAB detection kit (Ventana Medical Systems) according to manufacturer's instructions. Slides were counterstained with hematoxylin and coverslips were mounted with Permount (Fisher Scientific).

## Image acquisition and analysis

Images of spread spermatocytes were acquired on a Zeiss Axio Observer Z1 Marianas Workstation, equipped with an ORCA-Flash 4.0 camera, illuminated by an X-Cite 120 PC-Q light source, with 100× 1.4 NA oil immersion objective. Marianas Slidebook (Intelligent Imaging Innovations, Denver Colorado; version 5.0) software was used for acquisition. Whole histology slides were scanned and digitized with the Panoramic Flash Slide Scanner (3DHistech, Budapest, Hungary; version 3.2.0) with a 20× 0.8 NA objective (Carl Zeiss, Jena, Germany) and processed in CaseViewer (version 2.4). High-resolution histology images were acquired with a Zeiss Axio Imager (version 3.2) microscope using a 63× 1.4 NA oil immersion objective (Carl Zeiss, Jena, Germany). All images were processed in Fiji (version 1.54 g)[114]. To quantify foci of RPA, DMC1, or RAD51, staging of spermatocytes was assessed by SYCP3 staining (% of axes synapsed) and only foci colocalizing with

chromosome axis (SYCP3-signal) were manually counted. Statistical analysis was done with Graphpad Prism (version 9 or 10).

## Structured illumination microscopy

Spermatocytes from juvenile *Nbs1ΔB* homozygous and heterozygous littermates (13–15 dpp) were prepared for surface spreading and subsequent immunofluorescence as described previously[65,115]. Briefly, testes were dissected and placed in 1× PBS, pH 7.4 at room temperature before removal of the tunica albuginea. Seminiferous tubules were incubated in hypotonic extraction buffer (30 mM Tris-HCl pH 8.2, 50 mM sucrose, 17 mM trisodium citrate dihydrate, 5 mM EDTA, 2.5 mM DTT, 1 mM PMSF) for 20 min at room temperature. A homogeneous cell suspension was made in 100 mM sucrose and fixed on slides covered with a 1% PFA solution (pH 9.2) containing 0.15% Triton X-100 for at least 3 hr in a humid chamber. Slides were either used for immunofluorescence staining immediately or stored at −80 °C. For immunofluorescence, slides were blocked for 30 min at room temperature in PBST (1x phosphate-buffered saline (PBS) containing 0.1% Triton X-100) containing 3% nonfat milk. Slides were incubated with primary antibodies overnight in a humid chamber at 37 °C. Slides were washed 3 × 10 min in PBST, then incubated with secondary antibody for 1 hr at 37 °C in a humid chamber. Both primary and secondary antibodies were diluted in PBST containing 3% nonfat milk. After secondary antibody incubation, slides were washed 3 × 10 min in the dark with PBST and the slides were mounted with VECTASHIELD mounting medium containing DAPI (H-1000, Vector Laboratories).

Structured illumination microscopy (3D-SIM) was performed at the Bio-Imaging Resource Center in Rockefeller University using an OMX Blaze 3D-SIM super-resolution microscope (Applied Precision), equipped with 405 nm, 488 nm and 568 nm lasers, and 100× 1.40 NA UPLSAPO oil objective (Olympus). Image stacks of several μm thickness were taken with 0.125 μm optical section spacing and were reconstructed in Deltavision softWoRx 7.0.0 software with a Wiener filter of 0.002 using wavelength specific experimentally determined OTF functions. Maximum intensity projection images were acquired in Deltavision softWoRx 7.0.0 software. Data presented in this study were pooled from two independent staining experiments. To ensure correct alignment of fluorescent channels in all SIM experiments, multi-color fluorescent beads (TetraSpeck microspheres, 100 nm, Life Technologies, T7279) were imaged and used to correct shifts from passage of different fluorescence wave lengths through the optical path, as previously described[65]. SIM image analyzes were as previously described[65].

## Immunoblotting

About one million dissociated testis cells were boiled with NuPAGE LDS Sample Buffer (Life Technologies) and separated on 3%–8% Tris-acetate NuPAGE precast gels (Life Technologies) at 150 V for 70 min. Proteins were transferred to polyvinylidene difluoride (PVDF) membranes by wet transfer method in Tris-glycine-20% methanol, at 120 V for 40 min at 4 °C. Membranes were blocked with 5% non-fat milk in 1 × Tris buffered saline, 0.1% Tween (TBS-T) for 30 min at room temperature on an orbital shaker. Blocked membranes were incubated with primary antibodies overnight at 4 °C. Membranes were washed with TBS-T for 30 min at room temperature, then incubated with HRP-conjugated secondary antibodies for 1 hr at room temperature. Membranes were washed with TBS-T for 30 min and the signal was developed by ECL Prime (GE Healthcare) and scanned with the Image Lab (Bio-Rad; version 6.1.0 build 7). Primary and secondary antibodies are listed in Supplementary Data 2.

## SPO11-oligo complexes and denaturing PAGE

Because MRE11 deficiency appears to reduce or eliminate ATM activation in response to meiotic DSBs, we considered the *Atm*$^{-/-}$ mutant to be the most appropriate comparison for analysis of the sizes of SPO11 oligos in *Mre11-cKO*. Moreover, the greatly elevated numbers of

SPO11-oligo complexes in *Atm*$^{-/-}$[48] and *Mre11-cKO* (this study) allow us to measure SPO11 oligo sizes using samples from just two mice per genotype. By contrast, analysis of wild type requires many more mice[48]. To minimize unnecessary use of animals, we therefore decided to forego repeating a wild-type control (which has been shown previously[8,48,49]) in this study.

We purified SPO11-oligo complexes from testes of adults (>5 wk old) as previously described[113]. Briefly, testes were decapsulated, flash frozen in liquid nitrogen and stored at −80 °C. Testes were homogenized and extracts were cleared by ultracentrifugation. SPO11 was then immunoprecipitated with mouse monoclonal anti-SPO11-antibody 180 and protein A-agarose beads. SPO11 was eluted in SDS-containing buffer then subjected to a second round of immunoprecipitation with the same antibody. The material from the second immunoprecipitation was radiolabeled with terminal deoxynucleotidyl transferase and [α-$^{32}$P]-CTP then eluted with Laemmli sample buffer and separated by SDS-PAGE. Proteins were transferred to PVDF membrane. Alternatively, the size distribution of SPO11 oligos purified from two adult *Atm-KO* or *Mre11-cKO* mice was determined by protease digestion followed by radiolabeling with [α-$^{32}$P]-GTP followed by electrophoresis on a 15% denaturing polyacrylamide gel, which was then dried on filter paper as previously described[116]. Radio-labeled species were detected with Typhoon phosphor imager (version 4.0.0.4) after 48 hr exposure with Fuji phosphor screens, then quantified in Fiji (version 1.54 g)[114].

## Cell culture
Mouse embryonic fibroblasts (MEFs) were generated from E13.5 unsexed mouse embryos and cultured as described[23]. Mitotic index was quantified using early passage primary MEFs by measuring Ser10 phosphorylation of histone H3 by flow cytometry 1 hr after 3 Gy of IR exposure. Cells were treated with 10 μM of ATM inhibitor (KU55933, 11855, Sigma) 30 min before IR exposure as an ATM deficient control. Two independent MEF cell lines for either wild type or *Rad50-D69Y* were tested. For the colony formation assay, cells were immortalized as described[23], then plated with different doses of CPT for 24 hr and washed and cultured in drug-free media. For ATR inhibition, cells were treated with 50 nM of ATR inhibitor (VE822, S7102, Selleckchem) 2 hr before CPT treatment. After 10 days of growth, colonies were visualized by crystal violet stain (0.5% crystal violet, 25% methanol).

## Statistics and reproducibility
For immunoblotting and histology, experiments were performed with two independent biological replicates unless otherwise indicated. We selected and included representative images of immunoblots and histology sections in this paper. For labeling of SPO11-oligo complexes, experiments were performed twice independently, and both results were included either in main figures or in Supplementary Figs. The separation of SPO11 oligos by the denaturing PAGE was performed once. For immunostaining and SIM experiments, multiple images from two independent animal samples were captured and used for analysis. The exact number of cells in each quantification was indicated in figure legends.

When experiments were performed with two biological replicates, the quantification was plotted with mean ± range. In cases with more than three data points, error bars indicated the standard deviation.

Final figures were assembled using Adobe illustrator 2023 or 2024.

## Reporting summary
Further information on research design is available in the Nature Portfolio Reporting Summary linked to this article.

## Data availability
Raw and processed sequencing S1-seq, Exo7/T-seq, MRE11-ChIP, and SSDS data have been deposited in the Gene Expression Omnibus (GEO) repository under accession numbers GSE266258, GSE266151, and GSE266271. We used additional S1-seq and Exo7/T-seq data from GEO accession numbers GSE265863 and GSE229450[34,117]. We used SPO11-oligo data from GEO accession numbers GSE84689[37] and *Atm*$^{-/-}$ MRE11 ChIP-seq data from GEO accession GSE138915[14]. Code used for read processing and mapping is available online at https://github.com/yamadas2/mouse-S1seq, which also includes scripts for generating mean profiles and heatmaps around SPO11-oligo hotspot centers, and scatter plots to check correlation with SPO11-oligo maps and reproducibility between maps as previously described[34]. The mouse genome assembly mm10 (also known as GRCm38) is available at https://www.ncbi.nlm.nih.gov/datasets/genome/GCF_000001635.20/. Source data are provided with this paper.

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

## Acknowledgements

This article is subject to the Open Access to Publications policy of the Howard Hughes Medical Institute (HHMI). HHMI lab heads have previously granted a nonexclusive CC BY 4.0 license to the public and a sublicensable license to HHMI in their research articles. Pursuant to those licenses, the author-accepted manuscript of this article can be made freely available under a CC BY 4.0 license immediately upon publication. We thank W. Edelmann (Albert Einstein College of Medicine, New York, NY), R. Baer (Columbia University, New York, NY), Francesca Cole (MD Anderson Cancer Center, Houston, TX) and D. O. Ferguson (University of Michigan School of Medicine, Ann Arbor, MI) for generously sharing mouse lines. We thank A. Lukaszewicz (Univ. Michigan) and M. Jasin (MSK) for discussion. We thank the MSK Molecular Cytology core facility (N. Fan and M. Pulina) for histology; the MSK Integrated Genomics Operation (IGO) for sequencing; and the MSK Mouse Genetics Colony Management Group for assistance with mouse husbandry. MSK core facilities are supported by National Cancer Institute cancer center support grant P30 CA08748. The IGO was further funded by the Cycle for Survival and the Marie-Josée and Henry R. Kravis Center for Molecular Oncology. We thank M. Arter, E. Suranyi, H. Murakami, A. Shabro, V. Macera, and other members of the Keeney laboratory for discussions and experimental advice. We thank Cristina Madrid-Sandín (Universitat Autònoma de Barcelona) for assistance in generating one of the RPA2 SSDS maps from wild type. This work was supported by NIH grants R35 GM118092 (to S. Keeney), R01 GM59413 (to J.H.J.P.), R35 GM136278 (to J.H.J.P.), and the Brain Pool Program through the National Research Foundation of Korea (NRF) funded by the Ministry of Science and ICT (2022H1D3A2A01096332 to S. Kim).

## Author contributions

S. Keeney and S. Kim conceived the project. S. Kim, S.Y., T.L., J.H.K., M.M., J.X., and S. Keeney designed experiments. Specific experiments were performed as follows: Fig. 3e–g, J.X.; Fig. 5e-h and Supplementary Fig. 5e–i, M.M.; SIM and analysis (Fig. 6c, d and Supplementary Fig. 6b, c), T.L.; Supplementary Fig. 5a–c, J.H.K. All other experiments were performed by S. Kim. DNA sequencing data were analyzed by S. Kim and S.Y. S. Keeney contributed to analysis of all data. Some experimental animals were provided by C.C., D.Y.E., and L.F. S. Keeney, S. Kim, and J.H.J.P. supervised the research and secured funding. S. Kim and S. Keeney wrote the paper. All authors edited the manuscript.

## Competing interests

The authors declare no competing interests.
