## [Transparent Peer Review file · Nature Communications]

Mouse MRE11-RAD50-NBS1 is needed to start and extend meiotic DNA end resection

Corresponding Author: Dr Scott Keeney

Version 0:

Reviewer comments:

Reviewer #1

(Remarks to the Author)

The Mre11-Rad50-NBS1 is structurally and functionally conserved from yeast to mammals. Most of its functions have been first characterized in yeast such as its role in DNA end resection of DNA double strand breaks (DSBs). While this complex is only responsible for DNA end resection initiation and short-range resection in yeast, its role in mouse meiosis remained unexplored so far notably due to the lethality or strong phenotypes associated with the corresponding mutants. Thanks to the use of conditional mutants obtained by *in vivo* expression of the Cre recombinase, Kim et al report here that MRN is also responsible for the extension of DNA end resection in mouse meiosis. Overall, this study reveals slight but significant functional differences between mammals and yeast, and postulates that the nuclease activity is responsible for most of the long-range resection of meiotic DSBs in the mouse.

Overall, this manuscript contains a huge amount of work and results in a better understanding of the role of the MRN complex in meiotic DNA end resection in the mouse model, likely extrapolatable in human and slightly different from yeast.

1. As discussed at the end of the manuscript, MRE11 is essential for DSB formation in some organisms. It was so far unknown if this was the case in mouse. This paper now suggests that mouse MRE11 is dispensable for SPO11-DSB formation. This result may need to come first in the result section since the DSBs level in the MRE11-cKO will modulate the interpretation of the impact on resection.

In addition, as other conclusions in the paper, the dispensability of MRE11 for SPO11-DSB formation in mouse is toned down in the discussion section by the possibility that trace amounts of MRE11 in the MRE11-cKO context may be enough for DSB formation. This argument would be welcome in the result section, otherwise one wonders why the remarkable finding that MRE11 is dispensable for SPO11-DSB formation in mouse is not in the title of the paper.

2. SPO11-oligo immunoprecipitation does not measure DSBs per se, but only resected, endonucleolytically cleaved ends, and double-cuts. Because MRE11 regulates resection/endonuclease cutting, SPO11-oligo IPs may not be the appropriate methodology to quantify DSBs.

If MRE11-cKO (as the authors show) has a resection defect, there should be a decrease in “canonical” SPO11-oligos (“major species”), the ones released by endo cleavage. It is not clear whether this is the case on Fig. 2E and 2F.

The authors argue that the SPO11-oligo signal has a slower mobility in MRE11-cKO, which can perhaps be seen at lower exposure on Fig. S2b (thus should be referred to line 131). But the wild-type control is missing on denaturing PAGE fig. 2F, where the SPO11-oligos resulting from resection are formed and their size/frequency can be accurately estimated. The authors should therefore indicate the position of SPO11-oligos caused by endo cleavage/resection and double-cuts on Fig. 2F. It is not clear whether the main band just above 34nt corresponds to double-cuts or SPO11-oligos. It is referred as “major species” in the text, but it is important to clarify if these are SPO11 oligos (thus product of resection) or double-cuts (formed even in the absence of resection). This major band is present in ATM^{-/-}, that isn't resection deficient according to the authors on line 140 (or is it? further away it is mentioned that ATM^{-/-} has a resection defect, and the references 14 and 15 also point to a resection defect). Importantly, this major signal is also present in MRE11-cKO, albeit migrating slightly slower than in

ATM-/-.

Finally, the signal below 34nt that corresponds to smaller SPO11-oligo species (that are more abundant in the wild type and correspond to MRE11 dependent resection) should also be highlighted on the figure.

Altogether, the study on the MRE11-cKO is important and interesting, but an effort should be made to distinguish what is the product of resection and what is not on Fig 2F to help the reader. Whether ATM-/- has a (partial) resection defect or not, according to the literature, also needs to be clarified in the result section. Nevertheless, the total signal of SPO11 oligo IP is greatly elevated in the MRE11-cKO, in agreement with SPO11-DSBs being formed.

3. The main conclusion of this work is that MRN is responsible for both short- and long-range DNA end resection in mouse meiosis. This conclusion notably comes from the use of the Mre11-H129N nuclease dead mutant that shows reduced long-range resection. This conclusion assumes that the Mre11-H129N is a pure separation of function mutant, defective only for its nuclease activity. While it is discussed that this mutant seems to have normal ATM activity, is there definitive evidence that this mutant is fully functional in assembling the MRN complex and in all the other functions of the complex? If not, the authors may want to test this by Co-IP, for instance. Indeed, the use of the other hypomorphic MRN mutants in this manuscript shows that alterations of the MRN complex independently of the nuclease function can result in a resection defect. Additionally, although involving the nuclease activity of Mre11 is tempting, it is possible that the Mre11-H129N exerts a dominant negative effect on another resection factor.

4. The authors use the argument that the MRN complex mutants, the cKO and all the other hypomorphs, all being defective for long range resection is an indication that the MRN complex catalyzes long range resection. It has to be reminded here that the mre11 Δ mutant in yeast has a stronger resection defect than the mre11 nuclease mutant (reviewed in Cejka and Symington 2001), showing that it has a structural role promoting long-range resection independently of its nuclease activity. This point may be made clearer by the authors.

5. Despite the huge amount of work, it seems that this study misses definitive evidence supporting the involvement of the nuclease activity of MRE11 in long-range resection. This is indeed acknowledged by the authors themselves in the discussion starting line 435: while EXO1 is a major player in yeast, it could very well be that DNA2 took over the role of EXO1 in mammalian cells. So, it looks like the conclusion / title should be toned down accordingly.

6. Figures S2D-F show long-range resection in MRE11-cKO with a modal size of 897 nt. It is discussed that such resection could / likely comes from persisting wild type MRE11 in the cell so that resection can occur. However, under this scenario, as discussed for the MRE11-cHN mutant line 210, one would have expected to see an extent of resection similar to the wild type while it is significantly reduced. Could the authors comment on that?

7. The conditional depletion of MRE11 is a powerful tool with some drawbacks as discussed by the authors. For instance, the small enrichment only of unresected DSBs in MRE11-cHN compared to MRE11-cKO is explained by the putative persistence of wild type MRE11. This explanation is sound. However, what is striking in such a context is the strong resection phenotype since one would have expected at least some resection tracts as long as those from the wild type context. Could it be that the estimate of the resection tract length in MRE11-cHN is an over estimate of MRE11-HN tracts due to the contamination with "wt" tracts?

8. "Typo":

Figure 7A: the "RPM" indication is present for the WT but missing for all the mutants.

Reviewer #2

(Remarks to the Author)

Reviewer #3

(Remarks to the Author)

This manuscript focuses on the analysis of the initial step of DNA recombination—5' strand resection—during mouse meiosis. The authors establish several mouse models to investigate the role of the MRN complex in resection at SPO11-induced double-strand breaks (DSBs). Resection is tracked using sequencing methods that map DNA 3' ends after the degradation of single-stranded DNA (ssDNA), an established and reliable approach. All data presented are of high quality, and the limitations of the conditional models are appropriately discussed. The major findings regarding resection are as follows:

In the conditional knockout of MRE11, DSBs are not resected and form at elevated levels. This confirms the critical role of the MRN complex in resection and highlights its importance in regulating ATM, which in turn influences SPO11 activity. In

the conditional MRE11-H129N mutant, as well as in many additional MRN mutants, resection is impaired, suggesting that the MRN complex is necessary for generating approximately 1 kb of ssDNA, rather than just the initial few hundred base pairs. In summary, this work provides the most comprehensive characterization of the roles of the MRN complex and EXO1 in resection during meiosis within a mammalian system.

Questions:

1. Definition of initial and long-range resection:

The definitions of "initial" and "long-range" resection are unclear. Additional clarification is needed to explain what is meant by these terms in the context of this study.

From the perspective of this reviewer, initial resection refers to the resection activity carried out by MRX/MRN, whether it involves one incision or multiple incisions at a single DSB end. If the authors consider initial resection to be the closest incision/strand degradation by MRX/MRN at the DSB end, with all subsequent incisions classified as long-range resection, they should define it explicitly. However, this would be somewhat confusing. Additionally, if the authors define the first 200-300 bp of ssDNA generated by MRN as initial resection while subsequent 100-200 bp are considered long-range resection, it would be necessary to justify such a distinction. The mechanism would need to be different for the first few hundred base pairs compared to the following few hundred to justify such distinction.

Two models are proposed to explain the role of MRN in resection during mouse meiosis that would benefit from defining long-range resection:

First, the authors propose that MRN complex mediated resection is perhaps all that is needed in mice, and the only difference between yeast and mammals is that MRN can be more efficient in generating 1 kb of ssDNA than its yeast MRX counterpart. If this is the case, then the simple conclusion is that long-range resection is largely dispensable, with only a minor contribution from Exo1.

In second model, the MRN complex nicks the 5' strand, creating entry points for EXO1 and an unidentified second nuclease, thereby promoting long-range resection. If this is the case, it would mean that MRN indirectly facilitates extensive resection, which aligns with the current understanding of resection and there is nothing unexpected.

2. Figure 7A:

Figure 7A presents an interesting observation. In the central peaks reflecting ongoing recombination intermediate like D-loop, the top strand appears to have a much stronger signal in mutants. Does this suggest that only one end is resected and engages in recombination in mutants? Is such asynchrony of the central peaks observed at other hotspots?

Minor Comments:

• Line 98:

"These findings are markedly different from those in a recent Rad50 conditional knockout 20."

This sentence is vague, as the difference is not defined. Consider providing more context or removing it altogether.

• Figure 2F:

The Atm^{-/-} band should be shifted to the right to align with the lane.

• Stability of 3' Ends:

Are the 3' ends in wild type and mutants equally stable?

Reviewer #4

(Remarks to the Author)

The MRE11-RAD50-NBS1/Xrs2 complex has central roles in recombination, repair, and the DNA damage response. In the current manuscript, Kim and coworkers examine roles for the complex, in particular for MRE11, in the resection of meiotic double-strand breaks formed during mouse spermatogenesis. A spermatogenesis-specific MRE11 depletion mouse forms DSBs but fails to resect them, consistent with roles for MRE11, known from studies in budding yeast, in the removal of covalently-linked SPO11 and the initiation of resection. Somewhat surprisingly, a nuclease-dead allele of Mre11, at least in the context of the knockout model, can initiate resection with apparent efficiency, but displays substantially shorter resection tracts, as do several hypomorphic mutants in MRN components. These data argue persuasively for a role for MRN in the resection events that follow SPO11 removal, although, as authors acknowledge, precise interpretation is complicated by the possible involvement of residual wild-type protein in conditional knockout strains. Overall, the findings of this paper are quite novel and will be an important contribution to our understanding of meiotic recombination in mammals; it provides an alternate paradigm for MRN/X meiotic function that will be important in examining data from other species, as well. The manuscript is well-written and organized, and there is little to criticize. However, I have done my best to scour the manuscript for things to complain about, which are listed below:

1. While the nuclease-dead allele (Mre11-cHN) shows normal MRE11 protein levels, it would be useful to know if RAD50 and NBS1 protein levels are also normal. Similarly, interpretation of the hypomorph experiments might be clarified (or possibly complicated...) if one knew MRN protein levels in these mutants, as well.
2. For quantification of SPO11-oligos (Figure 2E, 4F), was the artifact signal removed? Or are these values just lane total?
3. For Figure 5D, it would be useful to report central/resection signal ratios, as in Figure 7C, to get a feeling for how much strand invasion is impaired by reduced resection (the impression is, not much). Was S1-seq done for Mre11-cHN? This would be useful for purposes of comparison.
4. Symbol use in Figure 7C is problematic; please consider enlarging them and using shapes more consistently (for example, circles for all and only wild-type).

Reviewer #5

(Remarks to the Author)

The paper by Kim et al. describes the characterization of germline-specific conditional knockout (cKO) mice of Mre11, which forms a complex with Rad50 and Nbs1 necessary for various DNA damage responses including meiotic recombination. In yeast, Mre11-Rad50-Xrs2(Nbs1) promotes the initiation of DNA double-strand breaks (DSBs) processing, but not for a long-range resection. By analyzing genome-wide ssDNA mapping in a Mre11 cKO and nuclease-dead mutants, the authors nicely showed that, in mouse male meiosis, Mre11 promotes not only the initiation of meiotic DSB end processing but also the long-range resection. This conclusion was further supported by the analysis of Rad50 and Nbs1 hypomorphic mutant mice as well as the double mutant of the Nbs1 with a defective allele of Exo1 (which is known to promote a long-range resection in yeast and mic). The double mutant of Nbs1dB Exo1-DA showed the additive reduction of ssDNA relative to either single mutant. The experiments were well-designed and performed with great care and most of the results are scientifically convincing and provide a new insight on the molecular mechanisms of DSB processing in mouse meiotic recombination. Moreover, the authors used these mutants to address some basic questions in the field such as the Rad51-Dmc1 localization. This paper provides a large number of data with high quality for the resection tract length in various MRN mutants. There are some concerns described below, which are welcome to address prior to the publication.

Major points:

1. For comparison between two genome-wide data by the S1 and ExoVII/ExoT mapping such as Figure 2A, how did the authors normalize the values between two mice without internal control (spiking)? It should be noted in the text and/or the legend. This is very important since the Mre11 cKO testes contained Mre11-positive and -negative spermatocytes.
2. In the second and third paragraphs of page 5, the authors explained the similarity of DSB phenotypes in the Mre11 cKO with the ATM mice by pointing out the double DSBs by Spo11. However, given that the cKO spermatocytes reduced Mre11 protein, it is possible that the Mre11 endonuclease and/or exonuclease activity enhanced by CtIP is attenuated (ATM activates CtIP by phosphorylation) in the mutant spermatocyte. As a result, a nick with Spo11 on the DSB end could accumulate, which would be released under the experimental conditions by the authors. In this scenario, ~10 bp periodicity would be generated by the periodicity of Mre11 nicking. When the authors could not deny this possibility, it is better to soften the idea of double-DSBs to explain the mutant mouse phenotypes.
3. The phenotype of Mre11-cHN, a phospho-di-esterase mutant, looks interesting. Given enough ssDNAs formed in the mutant, which would be sufficient for downstream recombination reaction, but the mutant spermatocytes did not finish recombination. This suggests a role of Mre11 after the ssDNA formation in the meiotic recombination. In this line, it is interesting to check the localization of other recombination proteins such as Dmc1 (as shown in Figure 5 for other mutants). Moreover, Msh4/5 and Mlh1/3 analysis would be interesting, but not essential.

Minor points:

1. Lane 94: Please explain what age of “young” mice (5 weeks) were used in the S1-seq analysis. Why the authors used 14.5 ddp mouse in wild-type control while 5 weeks for Mre11 cKO.
2. Line 110, MRN-dependent ATM activation: Is there any experimental evidence which support reduced ATM activation in Mre11-cKO spermatocyte such as gammaH2AX and/or Hormad1 phosphorylation (chromosome staining or western blotting).
3. Line 120: It is nice to show the position of subsidiary peaks in Figure 1G by either arrows or arrowheads.
4. Lines 126-127: As shown above, although the authors claimed that double-cut products by Spo11 are generated in the absence of “Mre11-nuclease activity” by citing two papers (ref.45 and 47). However, this is overstated. Since the two papers used sae2/com1 and rad50S mutants, which still retain some Mre11 nuclease activities at least in vitro.
5. Figure 2F: Lanes for ATM and Mre11-cKO are coming from the same gel or separate gels. If they are coming from the same gel, please remove a rectangle frame (a line between two lanes) from the Figure. Or when they are coming from two gels, please explain how they compared the densities in two samples (the gel did not show faster migration of the bands in Mre11-cKO relative to ATM; lane 138).
6. Lines 248-249, “with faster migrating species more depleted than slower migrating ones”: Without proper statistics, it is risky to insist on this. The authors may soften the claim. Moreover, if the authors compare the reduction from the peak/plateau, longer ssDNAs seem to be reduced in Mre11-HN relative to the control.
7. In Figures like Figure 6C and 5E, the S1-seq read counts show a periodicity. It would be nice for the authors to mention this and discuss it with a possibility such as nucleosome position etc.
8. Figure 6D: Please add a statistical comparison between two alleles.
9. Supplemental Figure 1E: A green arrowhead in the S1 treatment (box with dashed lines) should be “red” or removed. And a red arrowhead should be green or removed.

Version 1:

Reviewer comments:

Reviewer #1

(Remarks to the Author)

The authors appropriately addressed all my concerns.

Reviewer #2

(Remarks to the Author)

Reviewer #3

(Remarks to the Author)

The authors addressed all questions of this reviewer. It is an excellent work.

Reviewer #4

(Remarks to the Author)

Thank you for addressing all of my concerns in a satisfactory manner. I think that the revised manuscript is ready for publication.

Reviewer #5

(Remarks to the Author)

The authors properly responded to my comments and helped me understand the content by providing lots of text in the response. This is a very important paper, which should be published.

Minor comments:

1. Since the authors analyzed DSB-end resection only in spermatocytes in mice (not oocytes). The authors should mention spermatocytes or male meiosis somewhere in the abstract (or title).

We appreciate the reviewers' thoughtful and constructive feedback on our study. We are pleased that they found the work valuable and insightful. In response, we have addressed the feedback with edits to the manuscript text and/or figures as detailed below. Reviewer comments are in black text; responses are in red.

Reviewer #1 (Remarks to the Author)

The Mre11-Rad50-NBS1 is structurally and functionally conserved from yeast to mammals. Most of its functions have been first characterized in yeast such as its role in DNA end resection of DNA double strand breaks (DSBs). While this complex is only responsible for DNA end resection initiation and short-range resection in yeast, its role in mouse meiosis remained unexplored so far notably due to the lethality or strong phenotypes associated with the corresponding mutants. Thanks to the use of conditional mutants obtained by in vivo expression of the Cre recombinase, Kim et al report here that MRN is also responsible for the extension of DNA end resection in mouse meiosis. Overall, this study reveals slight but significant functional differences between mammals and yeast, and postulates that the nuclease activity is responsible for most of the long-range resection of meiotic DSBs in the mouse.

Overall, this manuscript contains a huge amount of work and results in a better understanding of the role of the MRN complex in meiotic DNA end resection in the mouse model, likely extrapolatable in human and slightly different from yeast.

We appreciate the reviewer's positive response.

1. As discussed at the end of the manuscript, MRE11 is essential for DSB formation in some organisms. It was so far unknown if this was the case in mouse. This paper now suggests that mouse MRE11 is dispensable for SPO11-DSB formation. This result may need to come first in the result section since the DSBs level in the MRE11-cKO will modulate the interpretation of the impact on resection.

In addition, as other conclusions in the paper, the dispensability of MRE11 for SPO11-DSB formation in mouse is toned down in the discussion section by the possibility that trace amounts of MRE11 in the MRE11-cKO context may be enough for DSB formation. This argument would be welcome in the result section, otherwise one wonders why the remarkable finding that MRE11 is dispensable for SPO11-DSB formation in mouse is not in the title of the paper.

We appreciate the reviewer's enthusiasm for the finding, but we prefer to keep the Results section focused on the things we can firmly conclude. Because we are working with a non-null mutation, our results are only suggestive at best about a possible requirement for MRE11 in DSB formation. We therefore consider it more appropriate to keep the analysis of this in the Discussion section, and to keep the Results focused on the much clearer resection phenotypes. To meet the reviewer partway, we added a sentence to the Results cross-referencing the Discussion (line 103). We are unsure what the reviewer means when saying that the DSB level in *Mre11-cKO* will "modulate the interpretation of the impact on resection." Given that the data clearly show that DSBs are made, we think the interpretation of the resection data stands on its own.

2. SPO11-oligo immunoprecipitation does not measure DSBs per se, but only resected, endonucleolytically cleaved ends, and double-cuts. Because MRE11 regulates resection/endonuclease cutting, SPO11-oligo IPs may not be the appropriate methodology to quantify DSBs.

We certainly agree that quantification of SPO11-oligo complexes does not measure DSBs per se, but we are unsure what might have given the impression that we were claiming that it does.

We are using SPO11-oligo complexes as a way to test the prediction that there is substantial double-cutting in *Mre11-cKO*. We are not using them to quantify DSBs.

If MRE11-cKO (as the authors show) has a resection defect, there should be a decrease in “canonical” SPO11-oligos (“major species”), the ones released by endo cleavage. It is not clear whether this is the case on Fig. 2E and 2F. The authors argue that the SPO11-oligo signal has a slower mobility in MRE11-cKO, which can perhaps be seen at lower exposure on Fig. S2b (thus should be referred to line 131).

We agree that a decrease in resection-generated SPO11 oligos is predicted, and this is indeed the argument we develop in the paper. We respectfully disagree that this conclusion is not clear from the data, however. To clarify, as suggested, we added mention of Supplementary Fig. S2b to the text (line 133). We also added a lighter exposure of the autorad to Fig. 2e (similar to Supplementary Fig. S2b).

But the wild-type control is missing on denaturing PAGE fig. 2F, where the SPO11-oligos resulting from resection are formed and their size/frequency can be accurately estimated. We agree that, in an ideal world, it would be nice to have the wild type control side by side. However, reproducing the wild type necessitates a large quantity of testis tissue from many mice. The increase in SPO11-oligo complexes in *Atm*^{-/-} and *Mre11-cKO* allows us to use fewer animals. We previously published data showing the size distribution of SPO11 oligos in wild type (Neale et al. *Nature* 2005; Lange et al. *Nature* 2011; Lange et al. *Cell* 2016), and in the 2011 paper we documented how the size distribution compares between wild type and *Atm*^{-/-}. Moreover, because MRE11 deficiency is expected to reduce or eliminate ATM activation (and we provide evidence that it does indeed do so), the most important point of comparison for this experiment is *Atm*^{-/-}, not wild type. Given that our previous publication (Lange et al. *Nature* 2011) provides a common reference point to wild type should a reader be interested to compare, we chose this way to do the experiment because we did not feel it was ethically justified to sacrifice so many mice for such a marginal gain in the presentation of the data. We added an explanation of this decision to Methods (lines 852-858).

The authors should therefore indicate the position of SPO11-oligos caused by endo cleavage/resection and double-cuts on Fig. 2F. It is not clear whether the main band just above 34nt corresponds to double-cuts or SPO11-oligos. It is referred as “major species” in the text, but it is important to clarify if these are SPO11 oligos (thus product of resection) or double-cuts (formed even in the absence of resection). This major band is present in *ATM*^{-/-}, that isn't resection deficient according to the authors on line 140 (or is it? further away it is mentioned that *ATM*^{-/-} has a resection defect, and the references 14 and 15 also point to a resection defect). Importantly, this major signal is also present in MRE11-cKO, albeit migrating slightly slower than in *ATM*^{-/-}.

Thank you for the suggestion. We revised Fig. 2f as suggested to label which parts of the lane profiles come from double cuts and which come from resection initiation. One point of clarification: all of these species are SPO11 oligos, regardless of how they arise.

The main band above 34 nt is much less abundant in wild type (Neale et al. *Nature* 2005; Lange et al. *Nature* 2011; Lange et al. *Cell* 2016), and several lines of evidence have shown that most or all of this is from double cutting (which does occur even in wild type) (Johnson et al. *Nature* 2021). This band (and the ladder of 10-bp periodicity above it) is present in *Atm*^{-/-} (as the reviewer notes) because this mutant has elevated double cutting, as shown previously in yeast and mice (Johnson et al. *Nature* 2021; Prieler et al. *Nature* 2021; Lukaszewicz et al. *Cell* 2021). To clarify, we edited the text to explain in more detail the known size distributions for SPO11 oligos arising from resection vs. double cutting (lines 137-151).

The reviewer seems to have misread our mention of resection in *Atm*^{-/-} mice. We did not state on line 140 of the original manuscript that resection is not deficient in this mutant. Instead, we stated that resection occurs, and highlighted that this is distinct from *Mre11*-cKO. *Atm*^{-/-} mice do indeed have a resection defect, specifically a small increase in unresected DSBs and a substantial change in the length of resection tracts. However, because resection still initiates at most DSBs in the *Atm*^{-/-} mutant, the important point at this juncture in the paper is that SPO11 oligos arise from both resection and double-cutting in *Atm*^{-/-}, but appear to be almost exclusively from double-cutting in *Mre11*-cKO. We amended the text on lines 143-144 to clarify.

Finally, the signal below 34nt that corresponds to smaller SPO11-oligo species (that are more abundant in the wild type and correspond to MRE11 dependent resection) should also be highlighted on the figure.

As suggested, we marked the expected positions of SPO11 oligos resulting from MRE11-dependent resection initiation as 'canonical resection' in Fig. 2f.

Altogether, the study on the MRE11-cKO is important and interesting, but an effort should be made to distinguish what is the product of resection and what is not on Fig 2F to help the reader. Whether *ATM*^{-/-} has a (partial) resection defect or not, according to the literature, also needs to be clarified in the result section. Nevertheless, the total signal of SPO11 oligo IP is greatly elevated in the MRE11-cKO, in agreement with SPO11-DSBs being formed.

We hope the edits noted above have clarified sufficiently.

3. The main conclusion of this work is that MRN is responsible for both short- and long-range DNA end resection in mouse meiosis. This conclusion notably comes from the use of the *Mre11*-H129N nuclease dead mutant that shows reduced long-range resection. This conclusion assumes that the *Mre11*-H129N is a pure separation of function mutant, defective only for its nuclease activity. While it is discussed that this mutant seems to have normal *ATM* activity, is there definitive evidence that this mutant is fully functional in assembling the MRN complex and in all the other functions of the complex? If not, the authors may want to test this by Co-IP, for instance.

Previous publications definitively showed that the human MRE11 mutant protein is nuclease dead but maintains the ability to form a complex with other components and to activate *ATM* (Lee et al. *J Biol Chem* 2013), and the integrity of the mouse mutant MRN complexes in cells was also demonstrated (Buis et al. *Cell* 2008). We had already included mention of this literature in the manuscript; we edited to try to make it clearer (now lines 212-214). We also confirmed normal testicular levels of RAD50 and NBS1 proteins in the *Mre11*-cHN mutant (see response to Reviewer 4, point #1) unlike in *Mre11*-cKO (Fig. 1c), so we do not consider it to be ethically justified to breed and sacrifice additional animals just to perform confirmatory coimmunoprecipitations.

Indeed, the use of the other hypomorphic MRN mutants in this manuscript shows that alterations of the MRN complex independently of the nuclease function can result in a resection defect. Additionally, although involving the nuclease activity of *Mre11* is tempting, it is possible that the *Mre11*-H129N exerts a dominant negative effect on another resection factor.

We had already provided extensive discourse on a possible nuclease-independent function of MRN in resection extension in the Discussion. To address the reviewer's comment, we softened the wording about the role of MRE11 nuclease activity (changing "reveals" to "suggests") in the Results (line 255).

4. The authors use the argument that the MRN complex mutants, the cKO and all the other

hypomorphs, all being defective for long range resection is an indication that the MRN complex catalyzes long range resection. It has to be reminded here that the *mre11* Δ mutant in yeast has a stronger resection defect than the *mre11* nuclease mutant (reviewed in Cejka and Symington 2001), showing that it has a structural role promoting long-range resection independently of its nuclease activity. This point may be made clearer by the authors.

We agree with the reviewer's sentiment, but it was not our intention to argue that MRN "catalyzes long range resection," at least not as the sole interpretation. Throughout the paper, we were careful to make a distinction between formal interpretations of genetic results (which by their nature do not allow conclusions about direct biochemical mechanisms) and discussions of possible mechanisms. For example, in the Results section on the MRN hypomorphs, we wrote (starting on current line 317), "These findings reinforce the conclusion that the MRN complex must be fully functional to support normal long-range resection, i.e., that MRN roles in DSB processing are not limited just to resection initiation." Similar wording was used in the Abstract, last paragraph of the Introduction, and elsewhere. This is a formal genetic interpretation based on the observed phenotypes, and is correct as such. Note that it draws no conclusions about a specific biochemical function. It was only in the Discussion that we proposed that MRN nuclease activity might directly catalyze the extension of resection tracts by iterative nicking, but we did so in the context of discussing the alternative that MRN might have a nuclease-independent role instead.

The only place where we think our wording could have been misconstrued as invoking a direct catalytic function was the title (see next comment). However, that was not our intention, as our title stated that the MRN complex (not the nuclease) was responsible for both types of resection. To clarify, we changed the title to emphasize the necessity for MRN and avoid implying a direct role of the nuclease.

5. Despite the huge amount of work, it seems that this study misses definitive evidence supporting the involvement of the nuclease activity of MRE11 in long-range resection. This is indeed acknowledged by the authors themselves in the discussion starting line 435: while EXO1 is a major player in yeast, it could very well be that DNA2 took over the role of EXO1 in mammalian cells. So, it looks like the conclusion / title should be toned down accordingly.

As noted in the preceding response, although we did not claim a specific direct function of MRE11 nuclease, we changed the title to avoid potential misreading. We agree that our study falls short of "definitive" evidence for involvement of the nuclease activity, but we did not claim that it does provide such evidence. Moreover, this is a general limitation of all genetic experiments, which are never capable of proving a direct biochemical function in vivo. As a point of comparison, essentially all of the reviewers' concerns here apply equally to the question of whether available genetic evidence proves that yeast *Mre11* nuclease directly clips *Spo11* from DSB ends in vivo.

6. Figures S2D-F show long-range resection in MRE11-cKO with a modal size of 897 nt. It is discussed that such resection could / likely comes from persisting wild type MRE11 in the cell so that resection can occur. However, under this scenario, as discussed for the MRE11-cHN mutant line 210, one would have expected to see an extent of resection similar to the wild type while it is significantly reduced. Could the authors comment on that?

We had already discussed the implications of the juvenile *Mre11-cKO* results (currently lines 152-164 in Results and lines 438-445 in Discussion). We added a sentence to the *Mre11-cHN* section to relate the logic there to interpretation of the juvenile *Mre11-cKO* results (lines 226-229).

7. The conditional depletion of MRE11 is a powerful tool with some drawbacks as discussed by the authors. For instance, the small enrichment only of unresected DSBs in MRE11-cHN compared to MRE11-cKO is explained by the putative persistence of wild type MRE11. This explanation is sound. However, what is striking in such a context is the strong resection phenotype since one would have expected at least some resection tracts as long as those from the wild type context.

We agree, as this was precisely the logic we spelled out in the introduction to this experiment (lines 220-229).

Could it be that the estimate of the resection tract length in MRE11-cHN is an over estimate of MRE11-HN tracts due to the contamination with “wt” tracts?

We are unsure what the reviewer has in mind here, as the relationship between this question and the initial part of the comment is unclear. The important and surprising result from this experiment is that resection tracts are shorter than normal in *Mre11-cHN*. It is certainly likely that there are some normal tracts in the data because the Cre excision efficiency is not 100%. If so, presence of these tracts would indeed mean that we are overestimating the average tract length in those *Mre11-cHN* cells that recombined the floxed allele. We suspect that such an effect is negligible in our data because the contribution of deletion-escapers was undetectable in *Mre11-cKO* (as noted on lines 206-208), and because there is not an obvious shoulder in the *Mre11-cHN* profile at the position of the peak in the wild-type profile (Fig. 4d). Nevertheless, even if there were a measurable effect from a subpopulation of phenotypically normal cells, it would only reinforce our main result because it would mean that the real resection defect is a bit stronger than it appears. Please note that, although we are documenting the magnitude of the *Mre11-cHN* resection defect, we draw no conclusions from the specific quantitative value. It is thus not obvious to us what the reviewer’s concern might be.

8. “Typo”:

Figure 7A: the “RPM” indication is present for the WT but missing for all the mutants.

This was not a typo. We chose not to repeat the text to avoid cluttering the figure. To clarify, we specified in the legends to Fig. 5b and 7a that the scale bars all represent the same RPM value.

Reviewer #2 (Remarks to the Author)

Reviewer #3 (Remarks to the Author)

This manuscript focuses on the analysis of the initial step of DNA recombination—5' strand resection—during mouse meiosis. The authors establish several mouse models to investigate the role of the MRN complex in resection at SPO11-induced double-strand breaks (DSBs). Resection is tracked using sequencing methods that map DNA 3' ends after the degradation of single-stranded DNA (ssDNA), an established and reliable approach. All data presented are of high quality, and the limitations of the conditional models are appropriately discussed. The major findings regarding resection are as follows:

In the conditional knockout of MRE11, DSBs are not resected and form at elevated levels. This confirms the critical role of the MRN complex in resection and highlights its importance in regulating ATM, which in turn influences SPO11 activity. In the conditional MRE11-H129N mutant, as well as in many additional MRN mutants, resection is impaired, suggesting that the MRN complex is necessary for generating approximately 1 kb of ssDNA, rather than just the initial few hundred base pairs. In summary, this work provides the most comprehensive characterization of the roles of the MRN complex and EXO1 in resection during meiosis within a mammalian system.

Thank you for the positive evaluation.

Questions:

1. Definition of initial and long-range resection:

The definitions of "initial" and "long-range" resection are unclear. Additional clarification is needed to explain what is meant by these terms in the context of this study.

From the perspective of this reviewer, initial resection refers to the resection activity carried out by MRX/MRN, whether it involves one incision or multiple incisions at a single DSB end. If the authors consider initial resection to be the closest incision/strand degradation by MRX/MRN at the DSB end, with all subsequent incisions classified as long-range resection, they should define it explicitly. However, this would be somewhat confusing. Additionally, if the authors define the first 200-300 bp of ssDNA generated by MRN as initial resection while subsequent 100-200 bp are considered long-range resection, it would be necessary to justify such a distinction. The mechanism would need to be different for the first few hundred base pairs compared to the following few hundred to justify such distinction.

Thank you for this comment, which prompted us to think more carefully about the wording we are using. The MRX part of meiotic resection in yeast seems to match well with "short-range" resection in vegetative cells. Because of this, and because meiotic resection involves a handoff to another nuclease that goes considerably further than MRX does (albeit less far than in vegetative cells), the yeast paradigm seems to partition resection into two steps that can be referred to as short-range (also defined as MRX-dependent) and long-range (also defined as being carried out by something other than MRX) by analogy with resection in vegetative cells. We are now finding that things are somewhat different in mouse, with MRN being necessary (directly or indirectly) for a substantially larger fraction of the total resection length and EXO1 contributing more of a "polishing" function. This strains the short-range/long-range terminology if the definition of these terms must include MRX/N dependency as well as resection tract length.

To address this issue, we edited the manuscript throughout to avoid ascribing a role for MRN in "long-range" resection, instead using wording that emphasizes its role in generating normal-length resection tracts and that draws a distinction with the "short-range" limitation of MRX-dependent resection in yeast. We mention "short-range" and "long-range" when introducing or discussing yeast resection, but we no longer use these terms to describe resection in mouse.

Two models are proposed to explain the role of MRN in resection during mouse meiosis that would benefit from defining long-range resection:

First, the authors propose that MRN complex mediated resection is perhaps all that is needed in mice, and the only difference between yeast and mammals is that MRN can be more efficient in generating 1 kb of ssDNA than its yeast MRX counterpart. If this is the case, then the simple conclusion is that long-range resection is largely dispensable, with only a minor contribution from Exo1.

In second model, the MRN complex nicks the 5' strand, creating entry points for EXO1 and an unidentified second nuclease, thereby promoting long-range resection. If this is the case, it would mean that MRN indirectly facilitates extensive resection, which aligns with the current understanding of resection and there is nothing unexpected.

As noted in the preceding response, we have removed mention of "long-range" resection entirely. We hope that this has made the presentation sufficiently clear.

We agree with the reviewer's summary of the second model, but we disagree that there would be "nothing unexpected" about it. The paradigm for meiosis so far is yeast, where it has been shown that Sgs1 (and by extension, Dna2) plays no discernible role whatsoever in normal meiotic resection. (It does play a modest role in the hyper resection that occurs in *dmc1* mutants, but that is not relevant to this discussion.) If the second model is the correct one, that would also be unlike the yeast paradigm, and thus we would consider it fair to call this "unexpected".

2. Figure 7A:

Figure 7A presents an interesting observation. In the central peaks reflecting ongoing recombination intermediate like D-loop, the top strand appears to have a much stronger signal in mutants. Does this suggest that only one end is resected and engages in recombination in mutants? Is such asynchrony of the central peaks observed at other hotspots?

Thanks for pointing this out, but this does not appear to be a reproducible feature. To more systematically evaluate the strand asymmetry of the central signal, we summed the reads from -100 to 1500 bp (bottom strand) or from -1500 to 100 bp (top strand) around the hotspot midpoint for each hotspot in multiple datasets (Review Fig. 1). Although the top-strand signal appears much stronger at the hotspot visualized in Fig. 7a, the summed read count is not reproducibly very asymmetric. The apparent asymmetry in the figure is probably mostly a visual artifact caused by the top strand reads being more clustered in a narrower region than the bottom strand reads are.

Minor Comments:

- Line 98:

"These findings are markedly different from those in a recent Rad50 conditional knockout 20." This sentence is vague, as the difference is not defined. Consider providing more context or removing it altogether.

Since we provide a more complete discussion of these differences in the Discussion section, we deleted the admittedly cryptic comment in Results.

- Figure 2F:

The Atm^{-/-} band should be shifted to the right to align with the lane.

Thank you, label fixed as suggested.

- Stability of 3' Ends:

Are the 3' ends in wild type and mutants equally stable?

The question is intriguing, but it is beyond the scope of what we can evaluate. Because our sequencing methods fully remove the ssDNA before ligation of the first adaptor, we don't have any direct information about the 3' ends. Since we observed no or only very small differences in the numbers of foci for DMC1 and RAD51 between wild type and MRN mutants with short resection, this may be indirect evidence that these mutants do not significantly affect the stability of 3' ends.

Reviewer #4 (Remarks to the Author)

The MRE11-RAD50-NBS1/Xrs2 complex has central roles in recombination, repair, and the DNA damage response. In the current manuscript, Kim and coworkers examine roles for the complex, in particular for MRE11, in the resection of meiotic double-strand breaks formed during mouse spermatogenesis. A spermatogenesis-specific MRE11 depletion mouse forms DSBs but fails to resect them, consistent with roles for MRE11, known from studies in budding yeast, in the removal of covalently-linked SPO11 and the initiation of resection. Somewhat surprisingly, a nuclease-dead allele of *Mre11*, at least in the context of the knockout model, can initiate resection with apparent efficiency, but displays substantially shorter resection tracts, as do several hypomorphic mutants in MRN components. These data argue persuasively for a role for MRN in the resection events that follow SPO11 removal, although, as authors acknowledge, precise interpretation is complicated by the possible involvement of residual wild-type protein in conditional knockout strains. Overall, the findings of this paper are quite novel and will be an important contribution to our understanding of meiotic recombination in mammals; it provides an alternate paradigm for MRN/X meiotic function that will be important in examining data from other species, as well.

The manuscript is well-written and organized, and there is little to criticize. However, I have done my best to scour the manuscript for things to complain about, which are listed below:

Thank you for the positive feedback. We're happy to address the comments, but certainly wouldn't have complained if there had been no scouring.

1. While the nuclease-dead allele (*Mre11*-cHN) shows normal MRE11 protein levels, it would be useful to know if RAD50 and NBS1 protein levels are also normal. Similarly, interpretation of they hypomorph experiments might be clarified (or possibly complicated...) if one know MRN protein levels in these mutants, as well.

The MRN hypomorphs used in this study have been extensively characterized in multiple publications by others. The protein level of MRE11-H129N was shown to remain unchanged compared to wild-type, along with RAD50 and NBS1 (Buis et al. *Cell*, 2008). We confirmed this (Review Fig. 2), but we elected not to include it in the paper because the RAD50 blot was not of sufficient quality in our opinion. Sacrificing more animals just to get a publication-ready figure when the main point is already well documented did not seem justifiable to us.

For *Mre11*-ATLD1, it was demonstrated that this mutation destabilizes the protein, leading to degradation of the other components (Theunissen et al. *Mol Cell*, 2003; Shull et al. *Genes Dev*, 2009). Similarly, *Nbs1* Δ B produces a truncated protein with reduced stability compared to full-length wild-type NBS1 (Williams et al. *Curr Biol*, 2002). The protein stability of *RAD50*-L1237F was previously analyzed (Al-Ahmadie et al. *Cancer Discov*, 2014), and the *RAD50*-D69Y

mutation is characterized in this study (Supplementary Fig. S5a). We edited the text to be sure the published analyses of protein stability are clearly cited (lines 300 and 304)

2. For quantification of SPO11-oligos (Figure 2E, 4F), was the artifact signal removed? Or are these values just lane total?

Thank you for highlighting the ambiguity. The artifact signal was excluded from the quantification of signal intensity, which was measured specifically from the region above the artifact band. This clarification has now been added to the figure legends for Fig. 2e and 4f.

3. For Figure 5D, it would be useful to report central/resection signal ratios, as in Figure 7C, to get a feeling for how much strand invasion is impaired by reduced resection (the impression is, not much). Was S1-seq done for *Mre11-cHN*? This would be useful for purposes of comparison. As suggested, we added the central/resection signal ratios for the other MRN hypomorphs. This information is most appropriately situated following the description of Fig. 7c, hence it is included as Supplementary Fig. S7d and described in the text that explains Fig. 7c (lines 402-412).

We did generate S1-seq libraries for *Mre11-cHN* (Review Fig. 3), but we opted to use the Exo7/T-seq data for this mutant because we didn't generate S1-seq libraries for age-matched wild type and heterozygote controls. The S1-seq showed an elevated ratio (0.53 and 0.59 from two biological replicates) compared to the ~0.4 typical of wild-type samples (see Fig. 7c). This elevation agrees with what we see in the Exo7/T-seq data (Fig. 4d). However, since we also provide evidence that there is a subpopulation of unresected DSBs in this mutant, it is misleading to compare this ratio directly to wild type or other mutants where the central signal is only from recombination intermediates. We therefore elected not to include this in the paper to avoid muddying the waters.

4. Symbol use in Figure 7C is problematic; please consider enlarging them and using shapes more consistently (for example, circles for all and only wild-type).

Thank you for the recommendation; the symbols in Fig. 7c have been revised for better consistency.

Reviewer #5 (Remarks to the Author):

The paper by Kim et al. describes the characterization of germline-specific conditional knockout (cKO) mice of Mre11, which forms a complex with Rad50 and Nbs1 necessary for various DNA damage responses including meiotic recombination. In yeast, Mre11-Rad50-Xrs2(Nbs1) promotes the initiation of DNA double-strand breaks (DSBs) processing, but not for a long-range resection. By analyzing genome-wide ssDNA mapping in a Mre11 cKO and nuclease-dead mutants, the authors nicely showed that, in mouse male meiosis, Mre11 promotes not only the initiation of meiotic DSB end processing but also the long-range resection. This conclusion was further supported by the analysis of Rad50 and Nbs1 hypomorphic mutant mice as well as the double mutant of the Nbs1 with a defective allele of Exo1 (which is known to promote a long-range resection in yeast and mic). The double mutant of Nbs1dB Exo1-DA showed the additive reduction of ssDNA relative to either single mutant. The experiments were well-designed and performed with great care and most of the results are scientifically convincing and provide a new insight on the molecular mechanisms of DSB processing in mouse meiotic recombination. Moreover, the authors used these mutants to address some basic questions in the field such as the Rad51-Dmc1 localization. This paper provides a large number of data with high quality for the resection tract length in various MRN mutants. There are some concerns described below, which are welcome to address prior to the publication.

Thank you for the positive evaluation and constructive suggestions.

Major points:

1. For comparison between two genome-wide data by the S1 and ExoVII/ExoT mapping such as Figure 2A, how did the authors normalize the values between two mice without internal control (spiking)? It should be noted in the text and/or the legend. This is very important since the Mre11 cKO testes contained Mre11-positive and -negative spermatocytes.

Thank you for the question; sorry if this was not completely clear. RPM values at each genomic position were averaged across datasets. For representative hotspot profiles (e.g., Fig. 1f), the averaged values were plotted after smoothing. For genome-wide average profiles (e.g., Fig. 2a), we co-oriented the averaged bottom-strand RPM values with top-strand values then averaged across all hotspots. For genotypes in which resection was initiated (i.e., all genotypes other than *Mre11-cKO*), we further normalized the averaged genome-wide profiles by setting the height of the resection peak equal to 1 after background subtraction. This normalization allows a clean comparison of the shapes of the resection tract distributions without confounding visual effects from differences in signal strength. These operations were already stated briefly in the figure legends and in more detail in Methods, but we edited both places to try to clarify further.

Unfortunately, we have not found a spike-in method that we consider reliable because of the variability in testis cellularity across different mutants. Nonetheless, we have found many features of the sequencing data to be highly reproducible, such that averaging datasets as described is well justified. We provide a detailed analysis of reproducibility in a separate study (Kim et al. *bioRxiv* 2024). Specifically, we find that spatial patterns for the distribution of resection endpoints and central signal and the ratio of central signal to resection signal are all extremely reproducible. This reproducibility arises in part because these are internal comparisons within a genomic map, so they are well-suited for RPM-based analysis. The absolute RPM value around hotspots is also reproducible, but somewhat less so because of differences in signal:noise ratio between sequencing libraries (i.e., differences in the amount of SPO11-independent background relative to SPO11-dependent signal) (see Fig. 2b for a visualization of this variability). Because of this variability, we generally avoid interpreting absolute RPM values and instead focus on the reproducible spatial patterns displayed using the

normalized average profiles. However, when differences in absolute RPM values exceed the typical variation we have documented for biological replicates, then we point such examples out (e.g., the discussion of the elevated total signal in *Mre11-cKO*; Fig. 2b). We added a paragraph to Methods (starting on line 638) to make these points more clear.

2. In the second and third paragraphs of page 5, the authors explained the similarity of DSB phenotypes in the *Mre11* cKO with the ATM mice by pointing out the double DSBs by Spo11. However, given that the cKO spermatocytes reduced *Mre11* protein, it is possible that the *Mre11* endonuclease and/or exonuclease activity enhanced by CtIP is attenuated (ATM activates CtIP by phosphorylation) in the mutant spermatocyte. As a result, a nick with Spo11 on the DSB end could accumulate, which would be released under the experimental conditions by the authors. In this scenario, ~10 bp periodicity would be generated by the periodicity of *Mre11* nicking. When the authors could not deny this possibility, it is better to soften the idea of double-DSBs to explain the mutant mouse phenotypes.

We certainly agree with the reviewer that *MRE11* nuclease activities are attenuated in the *Mre11-cKO* mice. That seems like the best explanation for the striking defect in resection initiation that we document. We believe we understand the basis for the reviewer's suggestion, but we respectfully disagree that this alternative explanation (periodicity in *MRE11* nicking) is plausible to explain our data, for multiple reasons. We apologize for the long response that follows, but we felt it necessary to reply at length since the reviewer has specifically asked us to explain why we think their model isn't possible. The bottom line is that, because every one of our findings reported here is perfectly consistent with and/or predicted from the well-established properties of SPO11 double cuts, and because many aspects of the data would be difficult or impossible to explain instead by a hypothetical periodicity in *MRE11* nicking, we stand by the conclusions in this part of the manuscript.

SPO11 double cuts have been definitively demonstrated by multiple lines of evidence in both yeast and mice, and they have been shown to occur in wild type in both species and to be elevated in *Atm* (*tel1*) mutants in both species as well (Johnson et al. *Nature* 2021; Prieler et al. *Nature* 2021; Lukaszewicz et al. *Cell* 2021). In yeast, it has also been shown that attenuating or eliminating Sae2-stimulated *MRE11* endo activity (e.g., in *sae2* or *rad50S* mutants) more or less completely eliminates canonical resection-generated Spo11 oligos while retaining Spo11 oligos that arise from double cuts.

These Spo11 double cuts show a remarkable nonrandom size distribution with predominant species beginning at roughly three helical DNA turns in length and incrementing in steps of ~10 bp. There is a plausible molecular model to explain both the minimum length and the 10 bp step size, and this model is supported by cryo-EM structures of yeast Spo11 bound to DNA (Yu et al. *NSMB* 2024) and by DNA binding properties of both mouse and yeast Spo11 proteins in vitro (Johnson et al. *Nature* 2021; Zheng et al. *bioRxiv* 2024).

Importantly, sequencing analysis in both yeast and mice has clearly demonstrated features of the periodically sized oligos that are well explained by double cutting but are inconsistent with canonical *MRE11*-dependent clipping of Spo11-bound DNA. For example, both ends of double-cut fragments in yeast have the pronounced sequence signature that is diagnostic of Spo11's preferred cutting sites, and this signature is markedly different from the sequence signature around the 3' ends of Spo11 oligos produced by canonical *Mre11* resection-initiation activity.

All of our findings reported here are perfectly consistent with the well-established properties of SPO11 double cuts.

The reviewer's alternative appears to allude to a model by Tamai et al. (*Nature Commun.* 2024), in which it is proposed that yeast MRX protein cuts adjacent to Spo11 at variable positions that coincidentally start at the same minimum size as Spo11 double cuts and then increment in steps of 10 bp in close register with the positions where double cuts are known to occur. The periodic Spo11 oligo sizes were observed by Tamai et al. with a *rad50* mutant (called *rad50-C47*) that was interpreted to be specifically defective for phospho-Sae2-stimulated MRX exonuclease activity, but proficient for endonuclease activity. In this model, the absence of exonuclease activity is necessary to reveal an otherwise cryptic periodicity in the spacing of Mre11 endonuclease nicking.

Several of our findings strongly argue against a similar interpretation for the *Mre11-cKO* mice. First, Tamai et al. explicitly proposed that their 10-bp ladder of Spo11 oligos arose from a situation where Mre11 endonuclease remains active but the exonuclease activity is (mostly) missing. Why would heavily depleting MRE11 protein in *Mre11-cKO* spermatocytes only affect the exonuclease activity? Put another way, if the SPO11 oligos we see are because of residual wild-type MRE11 endonuclease and not double cutting, why would that MRE11 protein be endo proficient but also completely exo deficient? If a fully wild-type MRN complex is present to clip the SPO11-bound strands, shouldn't it also be able to carry out its exonuclease function?

Second, we observed a massive increase in the amount of SPO11-oligo complexes (more than tenfold over wild type, similar to the increase we previously showed in *Atm*^{-/-}) (Fig. 2e) but less than a threefold increase in DSB ends detected by S1-seq (Fig. 2b). This is precisely what is predicted by our interpretation: If SPO11 cuts the same DNA molecule multiple times within a hotspot, it will release multiple SPO11 oligo complexes but the S1-seq (and Exo7/T-seq) libraries will pick up only the single outermost end on each side of a DSB cluster. By contrast, we are unable to think of a way in which essentially completely eliminating detectable MRE11 protein would increase the amount of SPO11 single cutting by more than tenfold while maintaining sufficient MRE11 endonuclease activity to process all of the extra DSBs. Both conditions would be necessary to explain why there are more total SPO11-oligo complexes without double cutting. But if there is more than tenfold more SPO11 single cutting, why do we not detect a commensurate increase in the S1-seq signal?

Third, there is a very close match of *Mre11-cKO* to *Atm*^{-/-} (similar fold increase in SPO11-oligo complexes; highly congruent size distributions of SPO11 oligos). There is definitive proof that the 10-bp Spo11-oligo ladder in yeast is mostly, if not entirely, from double cuts, and there is also compelling but less direct data that the same is true for *Atm*^{-/-} mice. The reviewer's alternative model would need to explain how you coincidentally get pretty much the same outcome from two completely different phenomena: double cutting in *Atm*^{-/-} but phased MRE11 nicking in *Mre11-cKO*. Such a coincidence would be hard to explain, but our model doesn't require any extra features because this is not a coincidence: the mutants look similar because they both cause an increase in the same phenomenon (SPO11 double cutting). Furthermore, the similar effects of both mutants fit well with a wealth of data about MRN-dependent activation of ATM in various cellular contexts and in different species, so our model is very parsimonious as a result.

3. The phenotype of *Mre11-cHN*, a phospho-di-esterase mutant, looks interesting. Given enough ssDNAs formed in the mutant, which would be sufficient for downstream recombination reaction, but the mutant spermatocytes did not finish recombination. This suggests a role of Mre11 after the ssDNA formation in the meiotic recombination. In this line, it is interesting to

check the localization of other recombination proteins such as Dmc1 (as shown in Figure 5 for other mutants. Moreover, Msh4/5 and Mlh1/3 analysis would be interesting, but not essential.) We may be misunderstanding, but the reviewer seems to be concluding that the *Mre11-cHN* mutant is unable to finish recombination. We are unsure what information is suggesting that, because we report on a failure to complete meiosis but do not have data concerning completion of recombination per se. However, the reviewer's comment does raise an interesting point, namely, that the meiotic defects of the *Mre11-cHN* mutant are worse than those in *Mre11-ATLD1* and *Nbs1 Δ B* despite *Mre11-cHN* having a longer average resection tract length. It is possible that the meiotic progression defects in *Mre11-cHN* are caused by defects in a downstream function as proposed by the reviewer. However, it is also possible that the progression defects are entirely the consequence of defects in resection and/or DSB formation. For example, we found that there were fewer SPO11-oligo complexes (Fig. 4f and Supplementary Fig. S4i) and reduced hotspot-associated sequencing signal (Supplementary Fig. S4j). As noted in the text, these results may indicate that there are fewer (processed) DSBs available to support recombination and chromosome pairing. We cannot rule out alternatives as mentioned in the text, but a reduction in the number of processed DSBs would readily explain the accumulation of leptotene and zygotene cells (due to delayed synapsis) and sporadic synapsis failure (affecting just a few chromosomes per cell) (Fig. 4m,n). Indeed, we previously showed (Kauppi et al. *Genes Dev* 2013) that reducing DSBs to about half the normal level gives a meiotic progression defect that appears quite similar to the one we document for *Mre11-cHN*.

We added discussion of this point (paragraph beginning on line 474). We agree with the reviewer that it would be interesting to have cytology data for recombination protein foci, but as this would necessitate a lengthy process of breeding and analyzing new experimental mice, we suggest that these experiments are outside the scope of the current paper.

Minor points:

1. Lane 94: Please explain what age of “young” mice (5 weeks) were used in the S1-seq analysis. Why the authors used 14.5 ddp mouse in wild-type control while 5 weeks for *Mre11* cKO.

The materials and methods section (“**Choice of sequencing method and animal ages**”) explains why we used animals of different ages. In brief, we preferred to use juvenile mice to reduce the background in S1-seq libraries, but for the conditional deletion mice using *Ngn3-Cre*, such as *Mre11-cKO* and *Mre11-cHN*, we needed to use older animals to reach sufficient depletion of MRE11 protein. Furthermore, we confirmed that the S1-seq profile is virtually identical in wild-type juvenile mice and adult mice (11 weeks old) (Kim et al. *BioRxiv*, 2024); therefore, we employed S1-seq from 14.5 ddp wild-type mice as a control here because it was the common control selected for S1-seq throughout the manuscript.

2. Line 110, MRN-dependent ATM activation: Is there any experimental evidence which support reduced ATM activation in *Mre11-cKO* spermatocyte such as gammaH2AX and/or Hormad1 phosphorylation (chromosome staining or western blotting).

The fact that ATM activation is impaired by *Mre11-cKO* is well characterized in *Mre11-cKO* MEFs (Buis et al. *Cell*, 2008) as well as in numerous papers demonstrating MRN-mediated ATM activation biochemically in vitro. In meiosis, we previously reported that *Mre11-ATLD1* and *Nbs1 Δ B* mutants show evidence of decreased ATM activity comparable to the decreases seen with these mutants in somatic cells (Pacheco et al. *PLoS Genet* 2015). Moreover, the fact that *Mre11-cKO* provides a striking phenocopy of *Atm*^{-/-} for numbers of SPO11-oligo complexes and size distributions of SPO11 oligos is well explained by the interpretation that ATM activation is defective in *Mre11-cKO*. Unfortunately, gammaH2AX is not a useful way to address this because other kinases such as ATR also contribute to gammaH2AX. Indeed, we observed

higher gammaH2AX staining in zygotene-like cells in *Mre11-cKO* (Review Fig. 4), which is reminiscent of *Atm* knockouts (Barchi et al. *Mol Cell Biol* 2005).

3. Line 120: It is nice to show the position of subsidiary peaks in Figure 1G by either arrows or arrowheads.

Fig. 1g and Supplementary Fig. S1c amended as suggested.

4. Lines 126-127: As shown above, although the authors claimed that double-cut products by Spo11 are generated in the absence of “Mre11-nuclease activity” by citing two papers (ref.45 and 47). However, this is overstated. Since the two papers used *sae2/com1* and *rad50S* mutants, which still retain some Mre11 nuclease activities at least in vitro.

We take the reviewer’s point that Spo11 double cutting has so far only been evaluated directly in *rad50S* and *sae2* mutants, so we amended the text to say this more explicitly: “double cutting in budding yeast can generate Spo11-oligo complexes when Mre11 nuclease activity is compromised by *rad50S* or *sae2Δ* mutations^{45,47}” (line 129-131).

Review Figure 4. Representative spreads of wild-type or *Mre11-cKO* spermatocytes showing SYCP3 (magenta) and γ H2AX (green) staining at the indicated stages or abnormal (“Zygonema-like”) cells with thickened and/or tangled axes.

5. Figure 2F: Lanes for ATM and *Mre11-cKO* are coming from the same gel or separate gels. If they are coming from the same gel, please remove a rectangle frame (a line between two lanes) from the Figure. Or when they are coming from two gels, please explain how they compared the densities in two samples (the gel did not show faster migration of the bands in *Mre11-cKO* relative to ATM; lane 138).

We had previously explained in the figure legend: 'Both samples were run on the same gel, but the intensity of the *Mre11-cKO* signal was increased approximately threefold to facilitate comparison.' The original gel image is shown in Review Fig. 5 and will be provided as Source Data so that readers will be able to judge for themselves. The purpose of this experiment is to analyze the size distribution of the oligos, not their absolute amount, so showing the gel lanes with matched intensities is the most appropriate way to display the data in our opinion. We also

had already explained in the legend that the lane profiles are background-subtracted and normalized to total lane signal.

We are unsure what the reviewer is referring to in the comment in parentheses. We didn't say that the bands in *Mre11-cKO* showed faster migration relative to *Atm-/-*, we said that *Mre11-cKO* had substantially less material migrating faster than the 34-nt marker, i.e., that there was a smaller AMOUNT of material in *Mre11-cKO* relative to *Atm-/-* in the part of the gel containing SPO11 oligos that arise from resection initiation instead of double cuts.

6. Lines 248-249, "with faster migrating species more depleted than slower migrating ones": Without proper statistics, it is risky to insist on this. The authors may soften the claim. Moreover, if the authors compare the reduction from the peak/plateau, longer ssDNAs seem to be reduced in *Mre11-HN* relative to the control.

We agree that we are unable to provide definitive statistical support. This is because we only have two replicates, which leaves this analysis statistically underpowered. Nevertheless, we note that the result is highly reproducible between the two replicates: the wild-type and *HN-het* samples are virtually identical to one another in both replicates, and the *cHN* samples show precisely the same difference from the other genotypes in both experiments (Fig. 4f and Supplementary Fig. S4i). Furthermore, the result would be statistically significant ($P = 0.0388$, t test) if we pooled the wild type and *HN-het* samples, suggesting that the lack of statistical significance is indeed likely to be solely because of insufficient statistical power. Because of the reproducibility and because this is a small point in the paper, we did not feel it was ethically justified to sacrifice additional mice just to achieve statistical power.

From the last sentence in the reviewer's comment, it appears that there may be a misunderstanding about the lane traces in Fig. 4g. If so, we apologize for the lack of clarity. The peak near the bottom of the trace (distance > 1.0) is mostly from the labeling artifact, not SPO11-oligo complexes, so it should be disregarded. What we are referring to as "faster migrating" species is the main bulk of the SPO11-oligo complexes in wild type, running at a distance > 0.5. The "slower migrating" species is the trailing signal above that (distance < 0.5). From the lane profiles (and the autorads themselves), we believe it is clear that the faster migrating species are highly depleted in *cHN* but the slower migrating species are hardly depleted at all (note the superimposable traces for distance < 0.5 in Fig. 4g).

To address these points, we softened the description as suggested (lines 264-266), we added a bar graph to Fig. 4g showing the quantification of the two replicate experiments, and we added explanatory labels to the lane traces in Fig. 4g to clarify which are the "fast" and "slow" migrating species we are referring to.

7. In Figures like Figure 6C and 5E, the S1-seq read counts show a periodicity. It would be nice for the authors to mention this and discuss it with a possibility such as nucleosome position etc. (The indicated figures are micrographs that do not contain S1-seq read count data. We are assuming for the response that the reviewer intended to refer to figures showing plots of S1-seq data at an individual hotspot, such as Fig. 5b.) The local peaks in the S1-seq profiles were analyzed in our previous study (Yamada et al. *Genes Dev*, 2020), which failed to identify any clear correlation with nucleosome positioning. We think that the peaks correspond to positions where resection preferentially stops, and we also think it likely that these positions may often include places where nucleosomes are present. The lack of a detectable correlation

between population average S1-seq peaks and nucleosome positions in spermatocytes (unlike in yeast) may be due to the variable nature of nucleosome positioning in mammalian cells. Because this has been presented in detail previously, we elected not to repeat this discussion in this manuscript.

Parenthetically, we note that the peaks do not appear to be periodic in nature: as far as we can tell, they do not tend to have a fixed distance between them, which would be required to conclude that they are periodic.

8. Figure 6D: Please add a statistical comparison between two alleles.

This was previously noted in the main text ($P = 2.2 \times 10^{-16}$, t test). We moved this to the figure, as suggested.

9. Supplemental Figure 1E: A green arrowhead in the S1 treatment (box with dashed lines) should be "red" or removed. And a red arrowhead should be green or removed.

Thanks for the suggestion; we modified the figure accordingly to avoid any confusion. We intended to indicate "either S1 or Exo7 treatment" for unresected DSBs.

REVIEWERS' COMMENTS

Reviewer #1 (Remarks to the Author):

The authors appropriately addressed all my concerns.

Response: Thank you

Reviewer #2 (Remarks to the Author):

Response: Thank you

Reviewer #3 (Remarks to the Author):

The authors addressed all questions of this reviewer. It is an excellent work.

Response: Thank you

Reviewer #4 (Remarks to the Author):

Thank you for addressing all of my concerns in a satisfactory manner. I think that the revised manuscript is ready for publication.

Response: Thank you

Reviewer #5 (Remarks to the Author):

The authors properly responded to my comments and helped me understand the content by providing lots of text in the response. This is a very important paper, which should be published.

Response: Thank you.

Minor comments:

1. Since the authors analyzed DSB-end resection only in spermatocytes in mice (not oocytes). The authors should mention spermatocytes or male meiosis somewhere in the abstract (or title).

Response: We added mention to the abstract that our experiments examined spermatogenesis, as requested.